# Convolutional networks for supervised mining of molecular patterns within cellular context

Irene de Teresa-Trueba [1,6,10], Sara K. Goetz [1,2,10], Alexander Mattausch [1,3], Frosina Stojanovska [1,2], Christian E. Zimmerli [1,7], Mauricio Toro-Nahuelpan [1,8], Dorothy W. C. Cheng[2,4], Fergus Tollervey[1,2], Constantin Pape [4,9], Martin Beck [1,4,7], Alba Diz-Muñoz[4], Anna Kreshuk [4], Julia Mahamid [1,4] ✉ & Judith B. Zaugg [1,5] ✉

Cryo-electron tomograms capture a wealth of structural information on the molecular constituents of cells and tissues. We present DeePiCt (deep picker in context), an open-source deep-learning framework for supervised segmentation and macromolecular complex localization in cryo-electron tomography. To train and benchmark DeePiCt on experimental data, we comprehensively annotated 20 tomograms of *Schizosaccharomyces pombe* for ribosomes, fatty acid synthases, membranes, nuclear pore complexes, organelles, and cytosol. By comparing DeePiCt to state-of-the-art approaches on this dataset, we show its unique ability to identify low-abundance and low-density complexes. We use DeePiCt to study compositionally distinct subpopulations of cellular ribosomes, with emphasis on their contextual association with mitochondria and the endoplasmic reticulum. Finally, applying pre-trained networks to a HeLa cell tomogram demonstrates that DeePiCt achieves high-quality predictions in unseen datasets from different biological species in a matter of minutes. The comprehensively annotated experimental data and pre-trained networks are provided for immediate use by the community.

Cryo-electron tomography (cryo-ET) produces three-dimensional (3D) snapshots of cellular landscapes at molecular resolution, making it possible to investigate structural and functional states of macromolecular complexes in their native environment, and to unveil how different macromolecular populations interact with cellular structures[1]. With improved instrumentation, sample preparation protocols, and automation, high-quality in-cell cryo-ET data are generated at increasingly high-throughput[2–4]. A prerequisite for subsequent structural analysis is the reliable identification of a relatively homogenous set of macromolecular complexes[5,6]. However, owing to the complex and crowded nature of the intracellular milieu, and limitations arising from cryo-ET image acquisition (low signal-to-noise ratio (SNR)

[1]Structural and Computational Biology Unit, European Molecular Biology Laboratory, Heidelberg, Germany. [2]Collaboration for Joint PhD Degree between EMBL and Heidelberg University, Faculty of Biosciences, Heidelberg, Germany. [3]Institute of Pharmacy and Molecular Biotechnology, Heidelberg University, Heidelberg, Germany. [4]Cell Biology and Biophysics Unit, European Molecular Biology Laboratory, Heidelberg, Germany. [5]Genome Biology Unit, European Molecular Biology Laboratory, Heidelberg, Germany. [6]Present address: Computer Science and Artificial Intelligence Lab, ENGIE Lab Crigen, Stains, France. [7]Present address: Department of Molecular Sociology, Max Planck Institute of Biophysics, Frankfurt, Germany. [8]Present address: Santiago GmbH & Co. KG, Willich, Germany. [9]Present address: Institute for Computer Science, Universität Göttingen, Göttingen, Germany. [10]These authors contributed equally: Irene de Teresa-Trueba, Sara K. Goetz. ✉e-mail: julia.mahamid@embl.de; judith.zaugg@embl.de

and incomplete angular sampling), such data mining of 3D cryo-ET volumes remains a major bottleneck.

A range of available semi-automated methods for segmentation of cellular structures and localization of macromolecular complexes (from here on, particles) in cryo-ET datasets are broadly classified as template-based and template-free approaches. Template Matching[7] is a commonly applied computational approach and is based on a point-wise numerical computation of a similarity coefficient (cross correlation) to a known template of the complex in question. It is accurate in the localization of large structures, but fails at identifying smaller or less dense particles[8], and is computationally intensive. Current template-free methods using classical image processing are designed for, and thus limited to, specific molecular configurations in which particles are associated with large cellular structures such as membranes or microtubules[9–11]. Both these methods typically require manual inspection and are therefore laborious and time-consuming. The advent of deep-learning methods, particularly convolutional neural networks (CNNs)[12], enabled more general and automated approaches for segmentation and particle localization in cryo-ET. The first of such approaches was a two-dimensional (2D) CNN that performs semantic segmentation of large structures, such as ribosomes or membranes[13]. However, its 2D nature makes it less suitable for particle localization, where probing the 3D structure becomes beneficial. More recently, DeepFinder, a fully supervised 3D CNN method that is based on the U-Net architecture[14] for multi-class semantic segmentation, has positioned itself as the state-of-the-art in automated particle localization in both simulated and real cryo-ET datasets[15,16]. Remaining limitations include the localization of less prevalent particles, and the interpretation of the obtained predictions within their cellular context.

Here, we present DeePiCt (deep picker in context), an open-source software that synergizes supervised convolutional networks for segmentation of cellular compartments (organelles or cytosol) and structures (membranes or cytoskeletal filaments), and localization of particles. We generated a set of comprehensively expert-annotated tomograms acquired on cryo-focused ion beam (cryo-FIB) lamellae of wild-type *S. pombe* for training and benchmarking of our method, which we openly provide to overcome the critical limitations arising from the absence of publicly available annotated experimental datasets. From here on, we refer to the term 'network' as the deep-learning algorithm itself and 'model' as the algorithm once already trained. We provide DeePiCt models, trained on experimental cryo-ET data, which show high data-mining performance and can be readily applied across species and datasets.

## Results

### DeePiCt for automated segmentation and particle localization
DeePiCt is based on deep-learning technologies. It combines a 2D CNN for segmentation of cellular compartments that are easily recognized in 2D, and a 3D CNN for particle localization and annotation of continuous structures, such as membranes and cytoskeletal filaments, that benefit from 3D information (Fig. 1). This synergy enables more precise particle picking and interpretation of their cellular context. The 2D and 3D CNNs are adapted from the original U-Net architecture[14] (Fig. 1a and Supplementary Note 1). U-Nets have become the standard for much of modern deep learning, with enormous success beyond segmentation, including in denoising[17] and reconstruction[18] methods for cryo-ET. Here, U-Nets are a natural choice for our dual purpose of addressing segmentation and detection (achieved through post-processing steps) since they involve less trainable parameters than architectures designed ad hoc for object detection[19]. The 2D CNN employs a fixed depth of 5 (4 max-pooling layers) and 16 initial filters (Fig. 1a). The 3D CNN allows multi-label learning (Supplementary Note 1) and adjustable architectural parameters for depth, number of initial filters, a batch normalization layer, and the dropout parameter in the encoder and decoder paths (Fig. 1a). These parameters can be set according to the

quality and amount of training data, and size, abundance, and shape complexity of the particle of interest. In general, larger particles benefit from larger depth to increase the receptive field of the network, and particles with a low-density print (low SNR with respect to the surrounding context) require more initial filters (Supplementary Table 1). The remaining optional layers (batch normalization and dropout) are well-known techniques in computer vision to avoid overfitting[20].

For training, each network requires tomograms and corresponding 3D binary segmentation masks of the structures of interest, for example organelle segmentations for the 2D U-Net and spheres representing particles for the 3D U-Net (Fig. 1b). The raw input tomograms are optionally pre-processed using an amplitude spectrum equalization filter to enhance image contrast (Fig. 1c and Extended Data Fig. 1). Both CNNs use the Adam optimizer algorithm and offer a choice of two loss functions (Dice and Generalized Dice loss) for training (Supplementary Note 1 and Supplementary Fig. 1). During training of the 2D network, tiles are randomly flipped and rotated in 90-degree increments to improve generalization. For the 3D CNN, we implemented a number of optional random transformations to the input images for data augmentation (Supplementary Note 1). In our experience, the 2D CNNs require about 6 fully segmented tomograms for training, while the 3D CNNs require about 5 tomograms for membrane segmentation, and a minimum of 300 annotated instances for particle learning independent of particle sparsity in the cellular volumes (Supplementary Fig. 2).

For predicting, the trained networks receive unseen tomograms pre-processed as the training data and output 3D probability maps that are subsequently automatically post-processed (Fig. 1d). Specifically, the predicted slices outputted by the 2D network are combined into a 3D volume, smoothened along the *z*-axis by applying a one-dimensional Gaussian filter, and thresholded (user-definable, default = 0.75) to generate a binary 3D map (Extended Data Fig. 2a–c). The output of the 3D CNN is thresholded at a user-defined probability value, followed by clustering, to generate a binary segmentation map. The clustered output can be integrated with contextual information from a binary map representing a tomographic region (for example, the cytosol segmentation from the 2D CNN) to reduce false positives. The mode of integration can be chosen among three different options: intersection, contact, or colocalization, depending on the users' specific application (Fig. 1e and Supplementary Fig. 3). For particle localization, a list of coordinates is generated from the clusters' centroids. For segmentations of continuous structures, such as cellular filaments, coordinates can be sampled along the segmentation centerline at a chosen spacing and exported; the particle orientations and structural features can then be obtained by subtomogram analysis in external software (for example, Warp[21], M[22], RELION[23], Dynamo[24], or EMAN2[25]). For more details on the method, including post-processing and performance evaluation (Supplementary Fig. 4), we refer to Supplementary Note 1 and our Github repository (https://github.com/ZauggGroup/DeePiCt).

### Generation of ground truth annotations in *S. pombe*
Developing and benchmarking deep neural networks require ground truth non-synthetic dataset. To this end, we created a comprehensive annotation of cellular features in tomograms acquired from wild-type *S. pombe* (Methods), representing diverse structures, particle sizes, and abundances.

We devised an iterative workflow combining template matching, DeePiCt, and manual picking, to annotate ribosomes, fatty acid synthases (FAS), membranes, organelles, and the cytoplasm in ten tomograms acquired by combining defocus and a Volta potential phase plate (VPP) and ten defocus-only tomograms (defocus) (Fig. 2 and Supplementary Tables 2–4). For annotating the nuclear pore complex (NPC), an additional dataset of 127 tomograms (denoted by defocus*) featuring ~354 NPCs[26] was used to ensure enough training data for this large, low abundance (on average three per tomogram), and

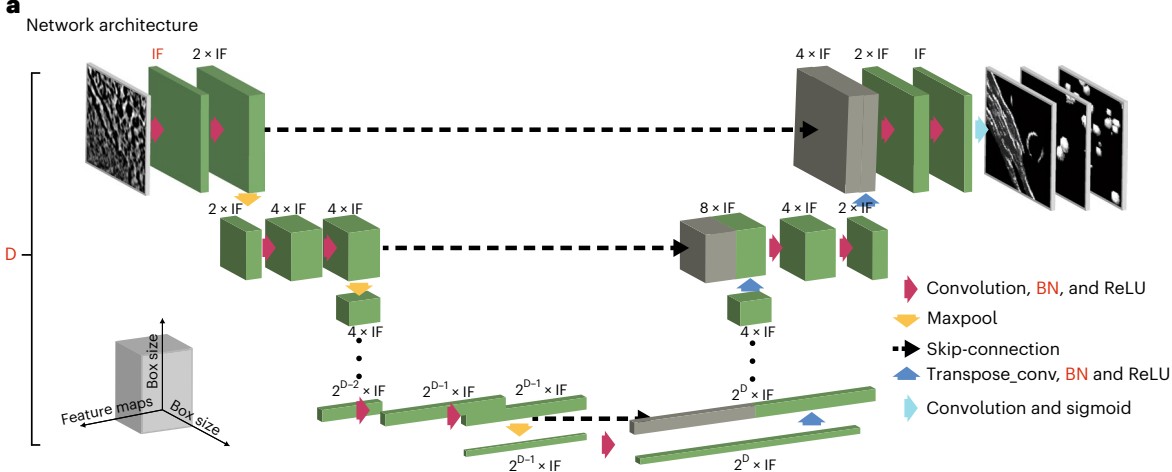

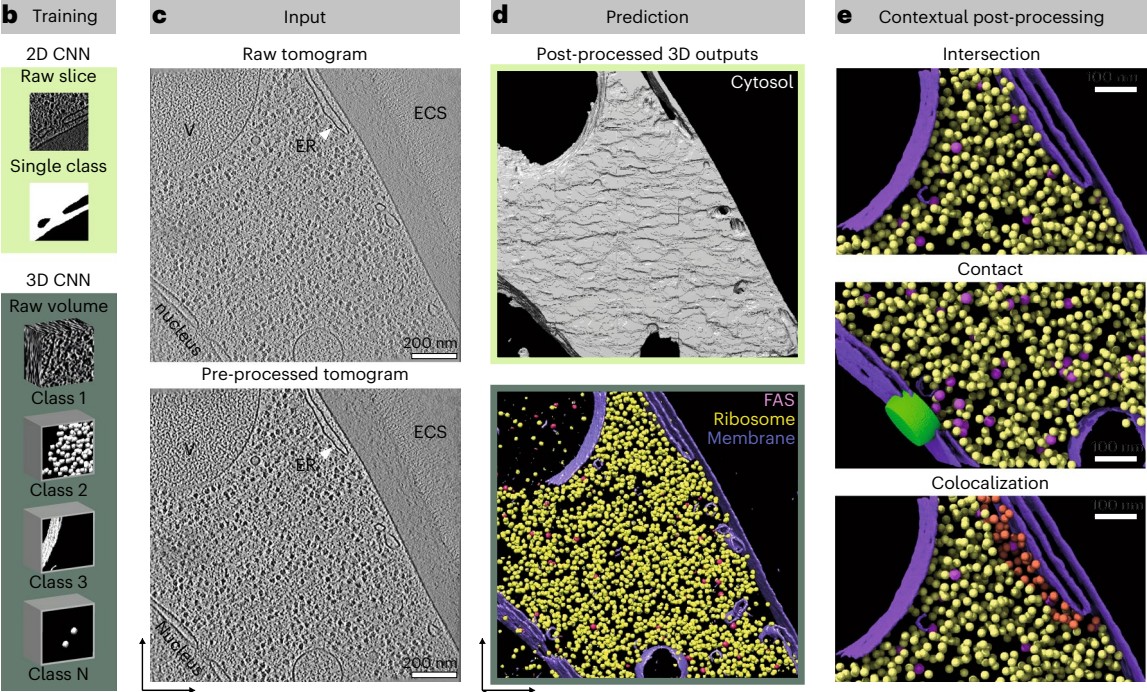

**Fig. 1 | DeePiCt 2D and 3D CNN architecture implemented in an automated workflow combining compartment segmentations and particle localizations in cryo-ET data. a**, CNN U-Net architecture: the 2D network performs all tensor operations on the 2D spatial coordinates, with depth (D) = 5 and number of initial features (IF) = 16; the 3D network performs tensor operations on the 3D spatial dimensions; architectural hyperparameters in red (D, IF, batch normalization layers (BN)) can be set by the user. **b**–**e**, The DeePiCt pipeline is used to train and predict various structural features in cryo-ET. **b**, The DeePiCt pipeline consists of two independent CNNs: a 2D network for compartment segmentation and a 3D network for particle localization. **c**, Trained networks are applied to input tomograms, which can be pre-processed with a spectrum-matching filter to improve image contrast. The example 2D tomographic slice visualizes the cytoplasm with the endoplasmic reticulum (ER), vacuoles (V), nucleus and extracellular space (ECS). **d**, DeePiCt raw predictions for cytosol (from the 2D CNN, top), membranes, ribosomes and fatty acid synthase (FAS; from the 3D CNN, bottom) are post-processed by thresholding, cluster size, and centroid fitting. **e**, The outputs of the two networks can be combined to include the cellular context by intersecting particle predictions with cytosol masks (top), selecting particles (NPC, green) in contact with specific membranes (nuclear envelope, purple, middle), and to identify particles (for example, ribosomes, orange) associated with specific organelles (for example, ER).

structurally flexible complex. For a detailed description of the ground truth construction see Methods.

In the ten VPP tomograms, we annotated a total of 25,311 ribosomes and 731 FAS. For the ribosomes, 61.6% (1,559 particles per tomogram (p.p.t.) on average) were found by initial template matching, 21% (541 p.p.t.) were added in iterative rounds of DeePiCt on the incomplete ground truth, and 17% (431 p.p.t.) by final manual annotations (Fig. 2j, Extended Data Fig. 3a,e, and Supplementary Tables 2 and 5). For FAS, since template matching failed, an initial round of manual picking contributed 58.96% (43 p.p.t.), followed by one round of DeePiCt on the incomplete ground truth, adding another 22% (16 p.p.t.). Additional rounds of DeePiCt were discarded since they led to a high false

positive rate. The final manual picking added a significant fraction of 19% (14 p.p.t.) (Fig. 2k, Extended Data Fig. 3c,f and Supplementary Tables 3 and 6).

In the ten defocus tomograms, we annotated 25,901 ribosomes and 366 FAS. Of the ribosomes, template matching found only 19% (498 p.p.t.), another 19% (502 p.p.t.) were added by DeePiCt trained on incomplete ground truth, and 61% (1,590 p.p.t.) were manually identified (Fig. 2j, Extended Data Fig. 3b,e, and Supplementary Tables 2 and 5). For FAS, 49% (18 p.p.t.) came from the initial manual annotations (template matching failed), 37% (14 p.p.t.) from DeePiCt trained on incomplete ground truth, and 14% (5 p.p.t.) from the final manual picking (Fig. 2k, Extended Data Fig. 3d,f, and Supplementary Tables 3 and 6). Overall, the final manual picking was crucial to generate a comprehensive ground truth annotation for FAS and, in defocus data, also for ribosomes, highlighting the value of our carefully curated annotations.

Total numbers of ribosomes in both acquisition types were comparable, while fewer FAS particles were detected in the defocus dataset likely owing to the lower SNR despite the application of an equalization filter (Extended Data Fig. 3, Supplementary Tables 2 and 3, and Supplementary Note 2). Nevertheless, our annotations of fully assembled FAS in both dataset are in agreement with levels of FAS-α and FAS-β quantified by previous proteomics studies[27,28] (Extended Data Fig. 4; Methods).

To assess the quality of the obtained ground truth annotations for ribosome and FAS, we performed structural 3D classification and refinement (Supplementary Note 2, Supplementary Figs. 5–7, and Supplementary Tables 7 and 8; Methods). Averages of ribosomes detected only by DeePiCt recapitulate 80S ribosomes, similar to the averages from all annotated ribosomes (Fig. 2l,m). 3D classifications of FAS demonstrated that DeePiCt together with manual annotations can recover these challenging shell-like structures independent of the data acquisition type (Fig. 2l,m, Supplementary Note 2 and Supplementary Fig. 6).

In conclusion, we provide comprehensive, high-quality annotations for large macromolecular complexes and cellular structures in *S. pombe* tomograms acquired with VPP and defocus. These set the ground for benchmarking the performance of the method and for future developments in particle detection approaches.

**Performance analysis and hyperparameter tuning of DeePiCt**
**Performance analysis of 2D CNN in VPP.** For assessing the performance of the 2D CNN, we evaluated two binary segmentation tasks on the ten ground truth VPP tomograms using fivefold cross-validation: segmentation of all organelles (all membrane-enclosed organelles and the nucleoplasm), and segmentation of the cytosol. The 2D CNN achieves high areas under the precision–recall curve (AUPRC; Supplementary Note 1 and Supplementary Fig. 4), with a median AUPRC of 0.92 for organelles and 0.98 for cytosol (Fig. 3a and Extended Data Fig. 5). Notably, basic hyperparameter tuning had close to no effect and a fixed architecture produced sufficiently good segmentations.

In addition, the 2D CNN has the potential to segment individual organelle types when provided with sufficient training data (Supplementary Table 4). Since it operates in 2D rather than 3D, it requires little memory (<1GB at the default batch size) and training (~15 min to train on 10 tomograms using a Nvidia V100 GPU; Methods).

**Hyperparameter tuning for the 3D CNN.** For the 3D CNN, we characterized the effect of the adjustable hyperparameters (Methods) using a cross-validation approach for ribosome, FAS, and membranes in the VPP dataset, and for NPC in the defocus* dataset.

To measure performance, we integrated the predictions of each target structure with an appropriate 'region mask' to remove false positives outside of expected regions (Supplementary Fig. 3): for ribosome and FAS, we used a cytosol prediction from the 2D CNN in 'intersection' mode; for membrane segmentation, we used the predicted cytosol from the 2D CNN in 'contact' mode; for NPCs, we used a 3D CNN prediction of the nuclear envelope in 'contact' mode.

Using this performance measure, we determined the hyperparameter combination that optimizes the task of localization or segmentation (Extended Data Fig. 6 and Supplementary Table 1). We found that the best hyperparameter setting is structure-dependent and related to particle size, density, symmetry, and abundance, among others, as well as on the receptive field of the network. Therefore, the architecture flexibility of our 3D CNN implementation is essential for segmentation of a diverse set of biological structures. Specifically, batch normalization shortens the learning time and notably improves performance for NPCs and FAS. Dropout layers and a number of data augmentation strategies did not improve performance (Supplementary Note 1 and Supplementary Fig. 2).

The physical receptive field of the network, that is, the context it sees for each prediction voxel, depends on the depth hyperparameter, and on the physical inter-voxel spacing. In all experiments, we used 4× binned tomograms with inter-voxel spacing of ~13.5 Å (Methods), sufficient for the detection of ~30-nm diameter particles investigated here. The binning can be optimized to detect different structures of interest. Lower binning retains higher-resolution information, but at the cost of higher noise and decreased physical receptive field of the network. The latter could be compensated with a higher depth, thus requiring more training data and possibly amplifying the well-known 'vanishing gradient effect' associated with deeper architectures[29]. Higher binning may be sufficient for larger structures such as organelles.

**Performance of DeePiCt in the same-domain setting.** Using the optimized networks, we analyzed the performance of the DeePiCt workflow in the same-domain setting (that is, training and testing in the same dataset type) using threefold cross-validation (Fig. 3b and Extended Data Fig. 7a–c). In VPP, it achieved a performance F1 score between 0.68 and 0.80 (median 0.79) for ribosomes, between 0.21 and 0.64 (0.46) for FAS, and a voxel-F1 between 0.58 and 0.90 (0.71) for membranes. For the NPC segmentation in defocus* (Fig. 3c), the performance depended on the quality of the tomograms, 23% of which were assessed to be high quality on the basis of lamella thickness and tilt-series alignment error (Methods; median voxel-F1 of 0.47 versus 0.19 for high and lower quality, respectively), even if the network was trained on the full dataset. Comparative analysis between DeePiCt single-class versus multi-label networks showed that single-class networks provide better results (Extended Data Fig. 6), even when employing loss functions that account for class imbalance, such as Generalized Dice, in multi-label networks training (Supplementary Fig. 1).

**Fig. 2 | Iterative comprehensive annotation of ground truth for non-synthetic data. a–h**, The three columns (left to right) show: the annotation process for ribosomes (**a**,**d**,**g**), FAS (**b**,**e**,**h**), and membranes (**c**,**f**). **a–f** Show the cumulative predictions of three rounds of DeePiCt for ribosomes and one round for FAS as cross-section overlaid on a representative tomographic slice (**a–c**; *n* = 20 tomograms) and in 3D view (**d–f**). Particles are classified as true positives recovered from the initial annotation (TPs recovered; yellow from template matching (TM) for ribosomes; green from manual annotation for FAS), new true positives provided by DeePiCt predictions (TPs new; blue), and false positives (FPs; red). **g**,**h**, 3D views of the resulting ground truth, including false negatives

(FNs) distinguishing unrecovered FNs from the initial annotations (pink) and final round of manual picking (salmon). **i**, Combined ground truth annotation of ribosomes, FAS, membranes, and NPC (green cylindrical mask, manual annotation). **j**,**k**, Relative contributions of the DeePiCt rounds for ribosome (**j**) and FAS (**k**) identification are plotted across ten VPP and ten defocus *S. pombe* tomograms: contribution of step 1 (yellow: TM and manual annotation for ribosomes; green: manual annotation for FAS; salmon: initial annotations not detected by DeePiCt), step 2 (blue); and step 3 (pink bars). **l**,**m**, Subtomogram averages of all ribosomes, exclusively DeePiCt-detected (TPs new; blue) and all FAS particles in VPP and defocus ground truth.

## Comparison of DeePiCt to state-of-the-art tools

We benchmarked DeePiCt against DeepFinder and template matching for localizing ribosomes and FAS in the VPP dataset (Fig. 3d,e).

Following the suggestions by the authors[15], DeepFinder was trained in a multi-class fashion and evaluated in the same threefold cross-validation setting as DeePiCt. In ribosome localization, DeepFinder performed

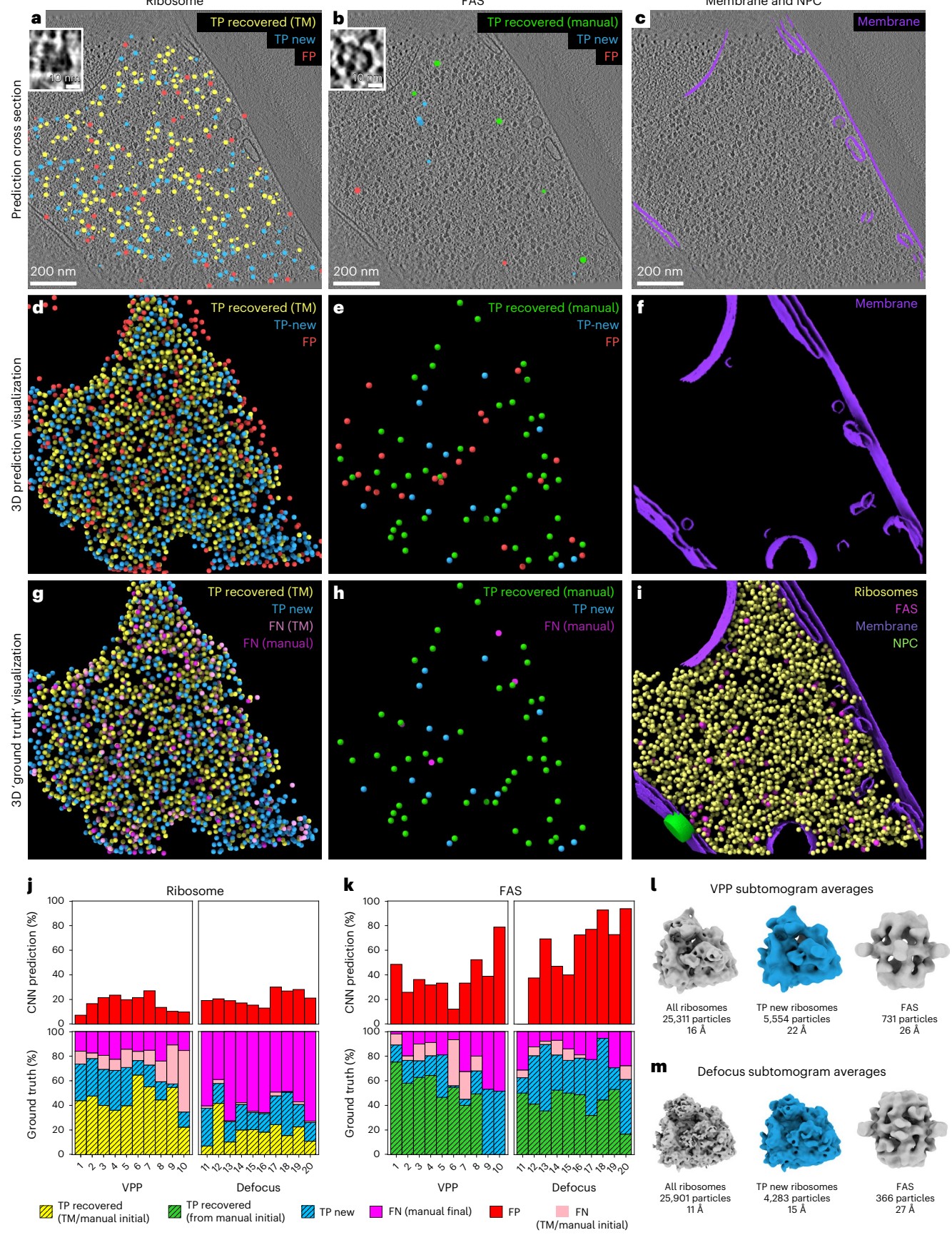

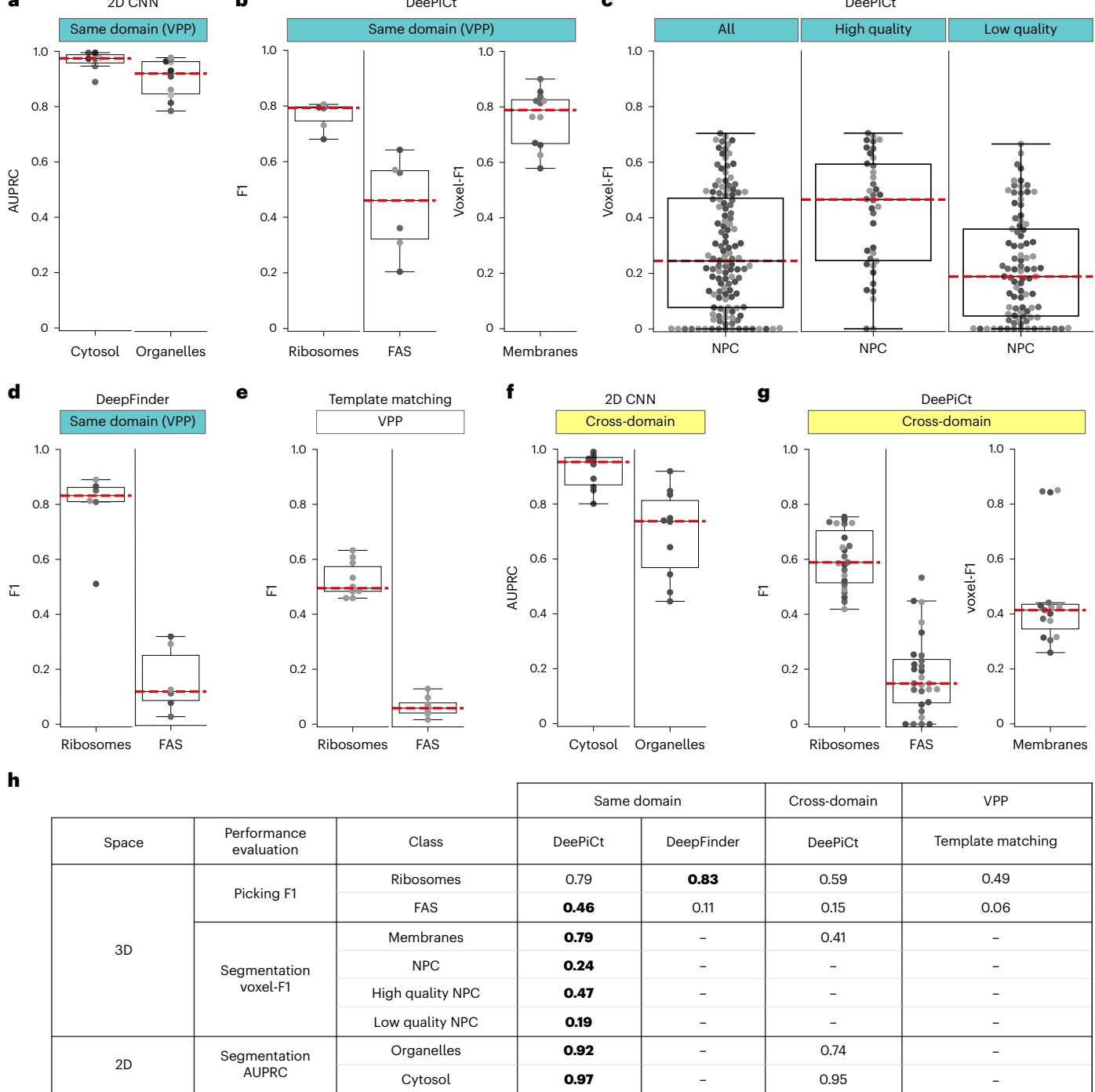

**Fig. 3 | DeePiCt performance, cross-domain generalization, and comparison with other methods. a**, Performance results of the 2D CNN for organelle and cytosol segmentation (*n* = 2 tomograms over 5 independent experiments) for training and testing in the same domain (VPP, cyan). The median AUPRC scores are indicated (red lines). **b**, Performance of DeePiCt for the same-domain setting (VPP, cyan) for ribosome localization, FAS localization, and membrane segmentation tasks (left to right), with *n* = 2, *n* = 2 and *n* = 4 tomograms, respectively, over 3 independent experiments. In each case, the corresponding architectures of the 3D CNN were optimized by hyperparameter tuning (Supplementary Table 1 and Extended Data Fig. 6). The median F1 score for ribosomes and FAS, and a median voxel-F1 for membranes are indicated (red lines). **c**, Same-domain NPC segmentation results (cyan). From left to right: performance in all (*n* = 42 tomograms over 3 independent experiments), high quality (*n* = 13), and lower quality defocus* tomograms (*n* = 29). **d,e**, DeepFinder (**d**) and template matching (**e**) particle localization results for ribosome and FAS.

Median F1 values are indicated (red line). **d**, DeepFinder results were achieved by training a multi-class DeepFinder network that simultaneously segments FAS and ribosomes (*n* = 2 tomograms over 3 independent experiments). **e**, Template matching results are shown for the VPP tomograms (*n* = 10 independent tomograms). **f,g**, Results of the cross-domain generalization (yellow) for the 2D CNN (**f**) and DeePiCt (**g**) (training on 10 VPP tomograms, testing on *n* = 10 defocus tomograms after amplitude spectrum equalization). Red lines indicate median performance values. Different shades of grey for dots correspond to cross-validation folds (DeePiCt and DeepFinder) or individual tomograms (template matching). Details of performance measurements (F1, voxel-F1, AUPRC) are described in Supplementary Note 1 (Supplementary Fig. 4). Box plot middle lines mark the median and the edges indicate the 25th and 75th percentiles; whiskers encompass all data that are not considered outliers (calculated by the Seaborn[51] box plot function). **h**, Median performance summary tables associated to results in plots **a**–**g**; bold numbers denote highest score values per class.

comparably to DeePiCt (median F1 0.83 versus 0.79). For FAS localization, DeepFinder performed significantly worse than DeePiCt (0.11 versus 0.46; *t*-test *P* = 0.007; Fig. 3d). Single-class DeepFinder networks for FAS failed to localize any particles. DeePiCt required ~17 h for training versus ~3 h for DeepFinder, while predicting and post-processing is equally fast for both (~500 clusters per minute; Methods). In contrast to DeepFinder, the trained networks of DeePiCt (models) for all structures mentioned in this work are open source and publicly available (Code Availability).

Template matching performance was measured by comparing the top 2,000 cross correlation peaks per tomogram of the raw output for ribosome and the top 1,000 peaks for FAS. It required several hours for detection in each dataset and showed worse performance than both deep-learning methods for the localization of ribosome, and completely failing for FAS (Fig. 3e).

### DeePiCt domain generalization across acquisition conditions

Defocus data has the advantage of yielding higher-resolution maps in subtomogram averaging compared to VPP[30]. Yet, it suffers from a lower image SNR and contrast, making particle localization and structure segmentation more challenging. Thus, we sought to assess the power of DeePiCt models trained on VPP for segmenting on defocus data.

The 2D CNN model trained on VPP tomograms shows slightly poorer performance on defocus data than in the same-domain setting (AUPRC 0.82 for cytosol and 0.42 for organelles; Fig. 3a,f). Similarly, DeePiCt 3D CNN performance dropped with respect to the same-domain setting to a median F1 of 0.59 for ribosome, 0.15 for FAS, and median voxel-F1 of 0.41 for membranes (Fig. 3g). In both networks, employing the spectrum equalization filter in the pre-processing step further improved performance in this domain generalization setting (Extended Data Figs. 5 and 7d–f). Furthermore, using the cytosol predictions as a 'region mask' improved the 3D CNN performance (Extended Data Fig. 7g,h). An example of DeePiCt segmentation results in the cross-domain setting is shown in Fig. 4a,b and visually resembles the ground truth annotation (Extended Data Fig. 8a).

### DeePiCt predictions result in high-quality subtomogram averages

To assess the quality of DeePiCt predictions on defocus data, we performed 3D structural classification on the output. For FAS, while DeePiCt identified fewer particles than in the ground truth (Supplementary Tables 8, 9), subtomogram averages revealed the two half domes of the barrel-shaped type I FAS with applied D₃ symmetry, consistent with the ground truth average and published structures from other yeast species (Fig. 4c,d and Extended Data Fig. 8b–e). We observed the phosphopantetheine transferase (PPT) domains required for activation of FAS along the equatorial plane and three additional equatorial densities that could not be assigned on the basis of published structures. Three densities inside each half dome connected to the central α wheel and in close proximity to the ketosynthase (KS) fit with the acyl carrier protein (ACP) of *Saccharomyces cerevisiae*[31] (Fig. 4d; Protein Data Bank (PDB) accession 2UV8) and *Pichia pastoris*[32] (Extended Data Fig. 8c; Electron Microscopy Data Bank (EMDB) accession EMD-12139). The ACP shuttles the growing acyl-chain between the different catalytic sites, and its localization has been suggested in connection with the activity of the whole multi-enzyme complex[31,33–36]. Here, maps of the *S. pombe* FAS complex in exponentially growing cells revealed a specific, native state, ACP site (Fig. 4d, Extended Data Fig. 8, and Supplementary Note 2).

For ribosomes, the particle averages and numbers derived from DeePiCt predictions are comparable to the ground truth annotations (Supplementary Tables 8 and 9, Supplementary Note 2, and Supplementary Figs. 7 and 8). 3D refinement of the ribosomal particles resulted in subtomogram averages with nominal resolutions of 11 Å (ground truth) and 15 Å (DeePiCt predictions) after multi-particle refinement in M[22] (Fig. 4e and Extended Data Fig. 8f). Hierarchical 3D

classification revealed a well-aligned class that was further refined in M to subnanometer resolutions of 9.3 Å (ground truth) and 9.4 Å (DeePiCt) (Fig. 4f–h and Extended Data Fig. 8i–k). This allowed the identification of tRNA occupying the P-site of the peptidyl transferase center (PTC) and the L1 stalk facing the E-site (Fig. 4g,h and Extended Data Fig. 8j, k).

### DeePiCt-predicted ribosomes reveal functional subpopulations

The large number of particles localized by DeePiCt in the defocus dataset in a high-throughput manner allows for examining subpopulations of functionally distinct complexes. Focused classification of all DeePiCt-predicted ribosomes on the head of the 40S small subunit revealed a subset with additional densities close to the head and at the exit tunnel (Fig. 4i and Supplementary Fig. 9). This class was also detected in the VPP and defocus ground truth datasets (Supplementary Tables 7–9), resolving densities for P- and E-site tRNAs in the latter (Extended Data Fig. 9a–e). The ribosome-bound ATPase eEF3 from *S. cerevisiae*[37] fitted well into the additional head density (CC 0.8972, EMDB accession EMD-12062; Fig. 4i and Extended Data Fig. 9a–c). During translation, this eukaryotic elongation factor facilitates binding of a new tRNA to the A-site via the ternary aminoacyl-tRNA–eEF1A–GTP complex[37]. Focused classification of all DeePiCt-predicted ribosomes at the ribosomal exit tunnel provided an average fitting the *S. cerevisiae* ribosome[38] (CC 0.9657, EMDB accession EMD-1667) with the rRNA expansion segment ES27L in a particular configuration[39] connecting to an additional density close to the ribosomal exit tunnel (CC 0.7938, PDB accession 3IZD; Fig. 4j and Supplementary Fig. 10). ES27L plays a role in translation fidelity. Enzymes, such as the methionine aminopeptidase (MetAP) that co-translationally processes the nascent peptide chain[40], nuclear export factor Arx1, which is released during ribosomal 60S maturation in *S. cerevisiae*[41], and its human homologue Ebp1, a translation regulator[42,43], bind at locations of the observed extra density. The binding factors recruit the flexible rRNA scaffold ES27L and cover the ribosomal exit tunnel with their MetAP-like folds. This structural class was also detected in the defocus and VPP ground truth datasets (Extended Data Fig. 9f–j and Supplementary Tables 7–9). Thus, the large number of particles obtained with DeePiCt predictions combined with structural analysis revealed functional subpopulations of ribosomes.

### DeePiCt reveals ribosome–mitochondria association

DeePiCt allows studying particle populations in specific contexts on the basis of their proximity to predicted organelles. We recovered cytosolic ribosomes within 25 nm distance to predicted ER and mitochondria from seven and three defocus tomograms, respectively. Subtomogram analyses for both subsets revealed one class with and one without a membrane density (Supplementary Figs. 11 and 12). Ribosomes showed a specific orientation relative to the membrane in the first case, while they were randomly orientated in the other. The latter likely arises owing to the highly crowded nature of the *S. pombe* cytoplasm (Fig. 4k,l), and, in the case of DeePiCt predictions, also from imperfect organelle segmentations (Fig. 4b). ER-bound ribosomes faced the membrane with the ribosomal exit tunnel in agreement with published structures from other species[38,44,45] (Extended Data Fig. 9k–n and Supplementary Tables 7–9). Mitochondria-bound ribosomes, which posed a particular challenge to structural analysis in previous studies[44,46], were found to interact with the membrane at an angular offset of around 35° in comparison to the ER-bound ribosomes (Fig. 4m,n, Extended Data Fig. 9o–r, and Supplementary Tables 7–9). Interestingly, a density connecting the ribosome to the mitochondrial membrane was detected on the large subunit in close proximity to the small subunit, but not to the ribosomal exit tunnel. It possibly represents the ribosome nascent chain complex in contact with the mitochondrion receptor OM14[47,48], which has yet to be structurally described. Thus, ribosomes close to mitochondria and ER exhibit

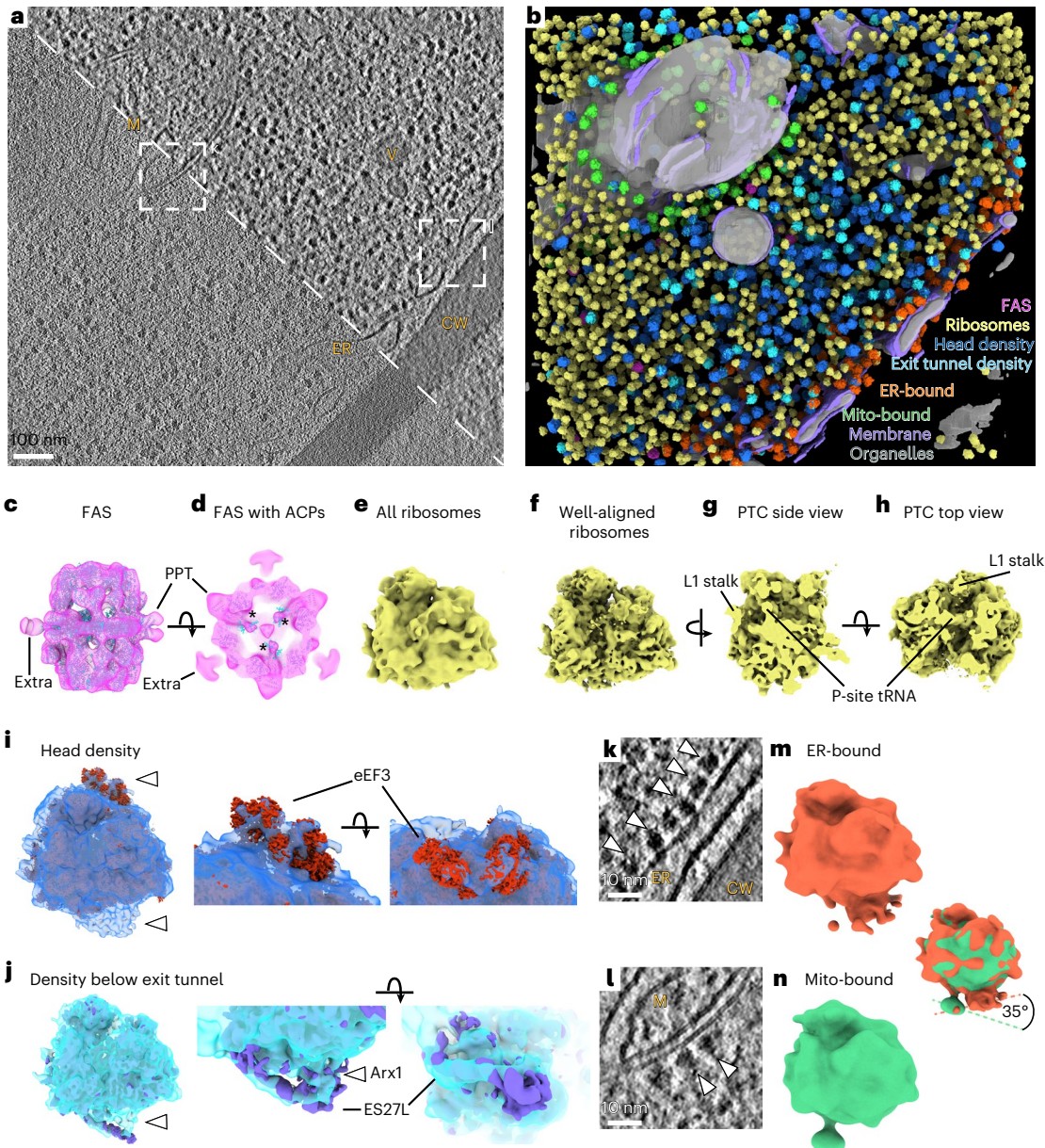

**Fig. 4 | DeePiCt enables exploration of macromolecular complexes in their cellular context. a**, 2D slice of a representative defocus *S. pombe* tomogram (*n* = 10 tomograms). Mitochondrion (M), vesicle (V), the ER and the cell wall (CW) are detectable in the raw (bottom left) and with improved contrast after amplitude spectrum equalization (top right). **b**, DeePiCt predictions generated with models trained in the VPP data. Organelles (gray), membranes (purple), FAS (pink, **c**,**d**), ribosomes (yellow, **e**–**h**) and subsets classified in RELION (head density, dark blue, **i**; exit tunnel density, bright blue, **j**), and within 25 nm of the ER (ER-bound, orange, **m**) and mitochondria (mito-bound, green, **n**). **c**, FAS subtomogram average (pink) fits the *S. cerevisiae* structure (cyan) including PPT domains. An extra density cannot be assigned. **d**, Cross-section of **c** close to the α-wheel with three densities fitting ACPs (asterisks). **e**, Subtomogram average of all ribosomes from ten defocus tomograms. **f**, Well-aligned ribosome subset detected by hierarchical 3D classification in RELION and refined in M. **g**,**h**, Slices through **f** reveal the PTC with a P-site tRNA and the L1 stalk facing

the E-site. **i**, Ribosome subclass with additional densities (white arrowheads) close to the head of the small ribosomal subunit, which fits eEF3 (red), and close to the ribosomal exit tunnel. **j**, Ribosomes classified for a density below the ribosomal exit tunnel (white arrowhead). The *S. cerevisiae* ribosome with ES27L in a particular configuration (left, purple) connects to the additional exit-tunnel density, which fits Arx1 bound to the 60S pre-ribosome (middle and right, purple). **k**,**l**, Different *z*-slices of the representative tomogram in **a** (white dashed boxes) show ribosomes bound to the ER (**k**) and a mitochondrion (**l**). **m**, An average of ER-bound ribosomes from seven tomograms shows a connection of the peptide exit tunnel of the large subunit to the membrane density. **n**, An average of mitochondria-bound ribosomes from three tomograms shows a linker connecting the large subunit, at a site close to the small subunit, to the membrane density. Overlay of the ribosomes in **m** and **n** shows different interfaces with the respective organelle membranes.

different interfaces with the respective membranes, potentially facilitating specific protein nascent chain membrane insertion or transfer into the organelle[38,47]. These results highlight the power of DeePiCt for high-throughput particle localization and cryo-ET segmentation to rapidly gain new biological insights.

## Trained networks can be readily applied to other species

As a demonstration of the domain generalization potential of our workflow across species and the ease of applying pre-trained models, we predicted ribosomes, membranes, and cytoskeletal filaments (actin and microtubules) in a published VPP HeLa cell dataset[45] (Fig. 5), for

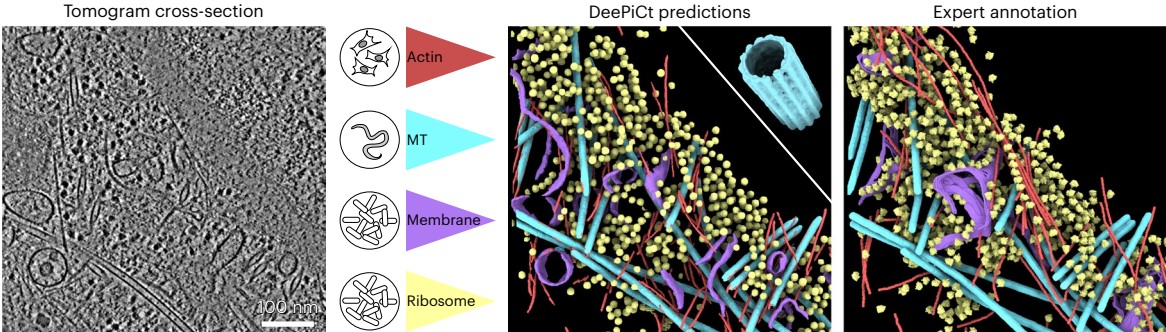

**Fig. 5 | DeePiCt generalization across species.** A dataset depicting a HeLa cell nuclear periphery (*n* = 1 tomogram)[45] is segmented by applying four independently trained DeePiCt networks. The results show the segmentation of actin filaments (red) trained on RPE-1 and MEF 3T3 tomograms, microtubules (MTs, cyan) trained on *C. elegans* tomograms, and cytosolic ribosomes (yellow) and membranes (purple) trained on *S. pombe* tomograms. The inset (top right) in the DeePiCt predictions panel shows the MT subtomogram average (cyan) obtained from the DeePiCt predictions.

which we manually generated a mask (using Amira[49]) of the cytoplasmic volume, to exclude the nucleus. For evaluation, we computed the voxel-F1 score using the publicly available annotations (EMDB accession EMD-11992 (ref. [45])). Using the ribosome and membrane networks trained on the ten VPP tomograms from *S. pombe* described above resulted in a voxel-F1 of 0.55 for ribosomes and a voxel-F1 of 0.18 for membranes (Extended Data Fig. 10a). However, the available expert membrane segmentation for the dataset covered only ER membranes. Visual inspection of the predicted membrane segmentation revealed a good fit (Fig. 5).

Additionally, we generated models for microtubules prediction, initially training a network for simultaneous actin and microtubule segmentations in four tomograms of Human retinal pigment epithelial-1 (RPE-1) cells[50]. Owing to the high preferential orientation of the cytoskeletal filaments in this data, the performance for microtubules segmentation was low (voxel-F1 0.26; Extended Data Fig. 10a,b). We therefore trained a second 3D CNN for microtubule segmentation on 11 tomograms from dissociated *C. elegans* cells, containing 890 microtubules in a wide range of orientations (Methods). This network achieved a voxel-F1 score of 0.83 in the HeLa cell tomogram (Extended Data Fig. 10a,c), exemplifying the importance of a training set with a high number of structures and orientations to mitigate the effect of the missing wedge on the prediction performance. These voxel-based DeePiCt predictions were then used to derive 3D coordinates for subsequent subtomogram averaging, revealing the 25 nm diameter hollow structure of HeLa cell microtubules, which is formed by 13 protofilaments (Methods; Fig. 5 and Extended Data Fig. 10c).

Finally, we trained a dedicated network for predicting actin filaments on five manually curated tomograms, two from RPE-1 cells and three from mouse embryonic fibroblasts (MEF) 3T3 cells, that contained approximately 3,740 actin filaments (Methods). This network shows a low voxel-F1 score of 0.10 in the HeLa cell tomogram (Fig. 5 and Extended Data Fig. 10a), which is likely caused by the finer structure of actin in comparison to microtubules, the minimal set of orientations sampled in the small number of training tomograms, and the fact that most training filaments are arranged in bundles, probably causing the CNN to learn this superstructure rather than patterns of individual filaments (Extended Data Fig. 10b).

Overall, these results show that the prediction of ribosomes and microtubules, representing large macromolecular complexes, is especially well-preserved across species and that the availability of diverse and well-annotated training datasets are crucial for good performance of DeePiCt. The results shown in Fig. 5 highlight the use of trained networks from DeePiCt to segment novel datasets spanning different species.

## Discussion

Our DeePiCt workflow facilitates accurate and fast localization of diverse structures in cryo-ET data of intact cells. The demonstrated high performance and the flexibility of the 3D CNN architecture offer a reliable tool for pattern recognition. This enabled us to detect lowly abundant particle species with a less dense structural signature (FAS) compared to ribosomes. The integration of structure segmentation (predicted by the 3D CNN) with the contextual information (predicted by the 2D CNN) excludes false positives in the particle localization and structures segmentation tasks and harnesses the cellular context to carry out spatial studies focused on regions of biological interest. This enabled us to investigate ribosomes in proximity to specific organelles (for example, ER versus mitochondria) and to obtain structural insights with functional implications. As the code is open source and Python-based, our flexible 3D architecture could further be expanded by implementing variations, such as ResNet encoders, atrous convolutions, class-normalization, positioning DeePiCt to serve as a tester for deep-learning techniques.

A major bottleneck in the field of supervised machine learning is the availability of expert curated training data. Here, we provide an experimental cryo-ET dataset of 20 *S. pombe* tomograms under two microscopy acquisition settings (VPP and defocus), together with high-quality comprehensive annotations of ribosomes and FAS, membranes, organelle, and cytosol segmentations. This constitutes the first gold-standard dataset in the field that is large enough for model training, which will enable benchmarking of current methods and spur the development of future computational tools for unbiased data mining in cryo-ET data. Subtomogram averaging of the annotated particles from either ground truth or DeePiCt predictions resulted in the first density maps of the *S. pombe* ribosome and fatty acid synthase, and further point to differences in the analysis of subtomograms from VPP or defocus tomograms, despite both being derived from wild-type *S. pombe* cryo-FIB lamellae.

The analysis of DeePiCt performance confirmed that data quality is important for its predictive ability. This was demonstrated specifically for predictions of the NPC, which, with its high degree of structural flexibility on the subunit and pore diameter levels inside cells[26], is a challenging target. High SNR and contrast are overall important for good performance during training and prediction with DeePiCt, exemplified by the higher performances for data acquired with a VPP. The introduction of a pre-processing equalizing filter improves the learning process during the training of 3D segmentation networks for particles with less dense print than the ribosome (for example, FAS), and especially for the generalization power across domains, including different microscopy acquisition conditions. For the 2D network, although pre-processing did not improve model performance on the

same-domain inference, it did improve cross-domain performance when training on VPP data and inferring on defocus data, or when training on both data types combined to segment organelles and cytosol. More elaborate tasks, such as prediction of individual organelle types, will likely require more training data or training of a dedicated 3D CNN with a tailored network architecture.

Our workflow allows easy adaptation to the segmentations of other structures, as demonstrated by the application of cytoskeleton segmentation networks. The networks show high-quality performance for microtubules in the HeLa cell dataset after training on data with broad orientation sampling of the filaments, producing segmentations that can be used for subsequent structural analysis. Actin predictions revealed a low F1 score and therefore likely require more training data, and sampling different orientations of the structural features, to improve performance. Altogether, the application of multiple segmentation networks to the HeLa cell dataset revealed that DeePiCt models trained on datasets from different microscopes, species, and conditions lead to reasonably good results in high-quality data. Although more in-depth analysis is needed to study the limitations for the applicability of DeePiCt networks on other datasets, the results presented here constitute the first step towards conducting large-scale quantitative analyses for structural biology using cryo-ET on cells from different laboratories and publicly available datasets. In this sense, the generated ground truth annotations in this study provide the community with a resource to improve and further develop cryo-ET object segmentation and detection tools, to ultimately enable broad exploration of particles in their cellular context. Together with the trained networks and the flexibility of the DeePiCt workflow, the software harbors great potential for quantitative cryo-ET studies in the future.

## Online content

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

## Methods

### Yeast cell culture

*S. pombe* K972 *Sp h-* wild-type haploid cells were recovered from frozen stock by streaking on YES agar plates (YES Broth, Formedium, 20 g agarose per liter) and incubated at 30 °C for 1–3 days. Colonies were restreaked on fresh YES agar plates and incubated 1–3 days at 30 °C. Single colonies were inoculated in 5 ml YES medium (YES Broth, Formedium, PCM0302, FM0618/8573) and grown at 30 °C, 170 r.p.m. overnight (NCU-Shaker mini, Benchmark). On the next day, cultures were grown to their log phase at an optical density at 600 nm of 0.5–0.6 and diluted beforehand in YES if necessary.

### Vitrification

Yeast cells were either diluted to optical density at 600 nm of 0.2–0.4 in YES medium or, following a wash step, in phosphate-buffered saline (PBS) containing 5% or 10% bovine serum albumin as external cryoprotectant. Transmission electron microscopy (TEM) grids (Quantifoil R1/2, Cu 200 mesh, holey carbon or SiO$_2$ film) were glow discharged on both sides for 45 s (Pelco Easy glow). Four microliters of the cell suspension were applied to the grids inside the chamber of a Leica EM GP (Leica Microsystems). Blotting from the back side of the support was performed for 1–2 s at 22 °C and 99% humidity. Grids were plunge frozen in liquid ethane cooled by liquid nitrogen and transferred into grid boxes until further usage.

### Cryo-FIB

TEM grids with vitrified yeast cells were clipped into an autogrid with a cut-out[52]. Mounted on a 45° pre-tilt shuttle, grids were transferred into an Aquilos Dual beam microscope (Thermo Fisher Scientific), sputter-coated with platinum for 10–15 s (1 kV, 10 mA, 10 Pa), and subsequently coated with organometallic platinum using the gas injection system (8 s with the stage positioned 3 mm below the coincidence point). In three independent sessions, three grids with five lamellae each were processed at a milling angle of 15°. Agglomerations of several cells were thinned in three steps of rough milling to a thickness of 5 μm at 1 nA ion beam current, 3 μm at 0.5 nA and 1 μm at 0.1 nA. Milling progress was visually monitored between each milling step with the scanning electron microscope beam (10 kV, 50 pA). Fine milling was performed at 50 pA to a target thickness of 200 nm. To render the lamellae conductive for TEM imaging, grids were sputtered with platinum for 5 s (1 kV, 10 mA, 10 Pa) and transferred into cryo-boxes.

### Cryo-ET

Autogrids with lamellae were loaded into a Titan Krios (Thermo Fisher Scientific) such that the axis of the pre-tilt introduced by FIB milling was aligned perpendicular to the tilt axis of the microscope[53]. Cryo-ET acquisition parameters are summarized in Supplementary Tables 7–9. Tomograms were acquired on a K2 Summit direct detection camera (Gatan) operating in dose fractionation mode utilizing a Quantum post-column energy filter operated at zero-loss (Gatan). A calibrated pixel size of 3.45 Å was used for the NPC defocus* data and 3.37 Å for the remaining datasets. Up to 14 tilt series were collected on a single lamella in low dose mode using SerialEM[54]. Starting from the lamella pre-tilt, images were acquired in 2° increments within a range of +58° to −40° using a dose-symmetric tilt scheme[55] with a constant electron dose per tilt image. For ground truth data, a set of ten tilt series were either collected with a 70 μm objective aperture or a VPP[56] (Thermo Fisher Scientific) with prior conditioning for 5 min. NPC data was collected as described in previous work[26,57], but at 1.5–4.5 μm defocus, with 3° increments and an effective tilt range of +50° to −50°.

### Tomogram reconstruction

Tilt movie frames were aligned using a SerialEM plugin. Tilt series were filtered according to the accumulated electron dose by Fourier cropping using the mtffilter function in etomo (IMOD/BETA4.10.12[58]), and

sorted by tilt angle using a python script. Four-times-binned tilt images were aligned in etomo (IMOD/4.9.4)[58] using patch tracking (typical residual error 0.291–0.569 pixels) and tomograms were reconstructed via weighted back projection. Tomogram thicknesses ranged between 80–310 nm.

### Ground truth annotation for organelles, cytoplasm, and membranes

In VPP and defocus datasets, organelles (mitochondria, vesicle, tube, ER, nuclear envelope, nucleus, vacuole, lipid droplet, Golgi apparatus, and vesicular body; Supplementary Table 4) and cytosol were annotated. Each compartment was identified through a unique numerical label to allow for selection of specific subsets of compartments. Segmentations of ten VPP tomograms were performed manually in Amira[49], and used to train a 2D CNN. Using this trained CNN, we predicted in ten defocus tomograms pre-processed with the spectrum equalization filter, and manually corrected the segmentations.

Membrane annotations were performed on ten VPP tomograms and five defocus tomograms. Initially, five VPP tomograms were annotated using Amira by manual segmentation on every two to three slices and subsequently interpolated. These were then used to train a membrane segmentation 3D CNN, whose predictions on the remaining five VPP and five defocus tomograms were manually corrected in Amira.

### Ground truth particle annotation in VPP data

Ribosome and FAS were localized in 4×-binned tomograms (13.48 Å voxel size) in an iterative workflow. Manually curated template matching for ribosomes and non-exhaustive manual picking for FAS (step 1; Supplementary Tables 2 and 3) were used for training 3D CNNs (step 2). CNN predictions were masked with a segmentation of the cytosol. For ribosomes, step 2 was repeated three times (always trained on combined predictions of step 1 and the preceding round) and each round consisted of three simultaneously trained networks with default hyperparameters (Supplementary Tables 2 and 3), except for the number of IF = 4, 8, and 32, to provide a cumulative prediction that is less overfitted to the incomplete training data. Cumulative predictions were manually revised in tom_chooser (ribosomes) and in EMAN2 (FAS) and manual picking was performed in either spectrum-matched or Gaussian-filtered tomograms ($\sigma = 3$) for up to three rounds in EMAN2 spt2_boxer[59] (step 3; Supplementary Tables 2 and 3). The particle lists were cleaned for duplicates by applying elliptic distance constraints to the coordinates (Supplementary Note 1 and Supplementary Tables 2 and 3).

Template matching for ribosomes was performed with pyTOM[60] by a 3D cross-correlation search over 1,944 Euler angle combinations using the large subunit of *S. cerevisiae* 80S ribosome map (EMDB accession EMD-3228 (ref. [61])) scaled to the corresponding pixel size, and a spherical mask (diameter 337 Å). The 2,000–3,000 highest cross correlation scores were manually revised in Gaussian-filtered tomograms ($\sigma = 3$) with tom_chooser (pyTOM toolbox[60]). For FAS in the VPP dataset template matching with a published *S. cerevisiae* FAS[33] map (EMDB accession EMD-1623) as reference failed.

### Ground truth particle annotation in defocus data

A procedure similar to the above was applied, except that: (i) the initial annotations for ribosomes were manually cleaned after template matching and initial FAS manual picking was incomplete; and (ii) in step 2 of the ribosomes ground truth construction, two of the three rounds of DeePiCt predictions were obtained using models trained on VPP data (first and second round of VPP ground truth construction; Supplementary Table 2).

### Comparison of cryo-ET-derived particle numbers with proteomics

Copy numbers of ribosomes and FAS per cell were calculated with the ground truth annotations for an *S. pombe* cell with the assumption of

30% cytosolic volume[62] of a total of 150 μm³ average cell volume[63,64], considering one fully assembled FAS complex is constituted by each six alpha and beta subunits.

## NPC manual localization

NPCs were manually localized as described previously[26]. Coordinates and initial orientations of 354 NPCs were manually determined in 127 4×-binned, SIRT-like filtered[58] defocus* tomograms[57]. The annotated NPC data was divided on the basis of quality criteria: 38 tomograms of quality 1 have a thickness below 300 nm, and a tilt-series alignment residual error below 0.7 pixels. Quality 0 was assigned to 89 tomograms with 300–395 nm thickness and a residual error of 0.7–5.0 pixels for tilt-series alignments.

## Voxel-level representation of ground truth

For ribosome and FAS, the lists of coordinates from the ground truth were used to paste spherical masks (with radii of 10.78 nm for ribosome and 13.48 nm for FAS; Code Availability). For the NPCs, a subunit mask obtained by prior 3D averaging in novaSTA (https://doi.org/10.5281/zenodo.3973623) was pasted at each of the eight subunit locations. The ribosome, FAS, membrane, organelle and cytosol masks in defocus and VPP along with the list of coordinates and respective tomograms are used in the subsequent training and performance analysis, and are available from the Electron Microscopy Public Image Archive (EMPIAR; Data Availability).

## Cytoskeletal filaments segmentation and subtomogram averaging

Microtubule and actin networks were trained on tomograms of Human retinal pigment epithelial-1 (RPE-1) cells generated in a previous study[50]. Segmentations of the individual cytoskeletal elements were performed using the filament tracing function in Amira[65,66], followed by manual curation. Trained networks were applied to a tomogram depicting the nuclear periphery of a HeLa cell and compared against the corresponding segmentations (EMDB accession EMD-11992).

For microtubules, an additional 3D CNN was trained on 11 tomograms containing 890 microtubules in a wide range of orientations from lamellae of cryo-FIB-milled *C. elegans* cells dissociated from a GFP::SPD-5, mCherry::histone worm line. This model was applied to the HeLa cell tomogram (EMDB accession EMD-11992) and the resulting predictions used to extract coordinates in the following steps: using skeleton3d[67] and a custom script in Matlab (version 2019a; Code Availability), particles were sampled along each filament at 6 pixel steps in a 4×-binned tomogram (corresponding to 101 Å with an unbinned pixel size of 4.21 Å), with each particle rotated by 360°/13 along the third Euler angle (second in-plane rotation). Particle cropping, alignment, and averaging were performed in Dynamo[24] (v.1.1.520). Particles were cropped from a 2×-binned tomogram with a pixel size of 8.42 Å per pixel, aligned over three rounds with a hollow tube as starting reference. Helical symmetry was applied starting from the second round of alignments. Resolution was determined at 39 Å (0.5 cut-off) by Fourier Shell Correlation (FSC) using the odd and even particles of the masked, final average.

## Subtomogram analysis for ribosomes and FAS

Contrast transfer function (CTF) estimations, generation of 3D CTF models and subtomograms were performed in Warp[21]. CTFs were first estimated in the sums of raw tilt movies and subsequently in the tilt series taking the tilt angles into account. Subtomograms containing ribosomes and their CTF models were reconstructed in volumes of 140³ pixels with a pixel size of 3.3702 Å and a particle diameter of 350 Å. Initial alignments were performed in RELION version 3.0.7[23] in 25 iterations of 3D classification, with the *S. cerevisiae* 80S ribosome (EMDB accession EMD-3228, low-pass filtered to 60 Å) as reference to generate an initial single-class average. 3D refinements were performed

with the resulting average as a reference. In defocus data, this average was further refined in M[22] to optimize particle poses, image and volume warping to model non-linear deformations. Particles were re-extracted and hierarchical 3D classifications (25 iterations each) were performed in RELION. For VPP data, 3D classifications were performed directly after 3D refinements in RELION. Focused classifications were performed with binary masks indicated in the respective figures.

FAS subtomograms and CTF models were reconstructed in cubic volumes of 160 pixels with a pixel size of 3.3702 Å and a particle diameter of 400 Å. Initial alignments were performed in RELION in 25 iterations of 3D classification into a single class, and refined with the 3D refinement option and applying $D_3$ symmetry using the *S. cerevisiae* FAS map (EMDB accession EMD-1623 (ref. [33]), low-pass filtered to 60 Å) as reference. Hierarchical 3D classifications (25 iterations each) were performed either after 3D refinements in RELION (VPP data) or after subtomogram re-extraction using the ribosome-optimized image and volume models in M (described above).

Final subtomogram averages of each particle class were obtained by 3D refinement and post-processing, filtered to their respective resolutions determined by FSC of two independently refined half maps at a cut-off of 0.143. Details of particle numbers and resolutions for each subtomogram average are summarized in Supplementary Tables 7–9. Visualization and calculation of cross-correlations (CC) between different maps and models, was performed with the UCSF ChimeraX package[68].

## CNN pre- and post-processing

Tomograms were first normalized to obtain uniform mean of 0 and variance of 1 in the frequency domain, before training. The spectrum equalization filter was then applied by matching the amplitude spectrum of each tomogram to the target spectrum of one manually selected high-contrast VPP tomogram (Tomogram TS_001; Extended Data Fig. 1b). Extraction of spectra amplitudes was done using fast Fourier transform followed by radial averaging of the amplitudes across the frequency domain. If the Nyquist frequency of the target tomogram is lower than that of the input tomogram, the target spectrum was padded with zeros to match the size of the input spectrum. Next, an equalization vector was created by dividing entry-wise the target spectrum by the respective input spectrum, converted into a rotational kernel and multiplied by the input tomograms in the frequency domain in combination with a sigmoidal-shaped low-pass filter to eliminate high-frequency noise. After back transformation, the tomogram exhibits a similar contrast to the target tomogram (Extended Data Fig. 1). For the 2D CNN, tomograms and training segmentations are processed slice-wise into 2D tiles with a fixed size of 288 × 288 pixels (256 × 256 pixels and 16 pixels padding on each side). For the 3D CNN, tomograms are by default split into cubic patches of 64 × 64 × 64 voxels, and 12 voxels overlap in each dimension.

Post-processing for the 2D network assembles the per-slice prediction into a 3D segmentation. Predicted tile segmentations are cropped on each side by 48 pixels to reduce artifacts around the edges, followed by reassembly into 3D stacks, with remaining overlapping areas averaged. A one-dimensional Gaussian filter is applied along the z-axis to reduce single-slice false positives (Extended Data Fig. 2).

For the 3D CNN, individual 64 × 64 × 64 voxel patches are reassembled into the probability map and the thresholded map (usually at threshold value of 0.5) subsequently clustered. Clusters can be filtered for size and context (within or close to a given organelle/cytosol segmentation output by the 2D network) for the final prediction map (Supplementary Note 1). A list of coordinates of cluster-centroids, representing the particle location predictions, is then exported.

## Evaluation metrics

To evaluate the particle localization task (ribosome and FAS), we defined true positives as those predicted particles whose coordinates

overlap with a ground truth particle within a tolerance radius (10 voxels, 135 Å), and reported the F1 score as the harmonic mean between recall (proportion of ground truth particles recovered) and precision (proportion of predicted particles that were true positives; Supplementary Note 1 and Supplementary Table 1). For the structure segmentation task (membrane or the NPC segmentation), we compared the ground truth masks with the predicted post-processed segmentation by calculating their voxel-wise precision and recall and reported the corresponding voxel-based F1 (voxel-F1) score, also known as Sørensen–Dice coefficient (Supplementary Note 1 and Supplementary Fig. 4).

## Cross validation and performance evaluation

For ribosome and FAS localization, and for membrane segmentation, a non-standard threefold cross-validation scheme where three (as opposed to five) subsets of eight VPP tomograms were used for training, and two for testing. In the test for domain generalization, the same three networks (trained on the eight VPP tomograms) were applied to the ten defocus tomograms.

The NPC predictions, treated as segmentation tasks, were evaluated in the defocus* independent data selected for that purpose using a threefold cross-validation where the total 127 tomograms were split into three random subsets with roughly the same number of tomograms each. Each fold consists of two such sets as training data and the remaining one for testing (Supplementary Note 1 and Extended Data Fig. 6d–f). The nuclear envelope predictions, which were used as 'region mask' in the 'contact' mode during post-processing of the NPC prediction, were achieved with a 3D CNN trained on 18 manually annotated tomograms, and which were uniformly distributed across the three dataset splits.

For organelle and cytosol segmentation evaluation, model performance was evaluated on the voxel-level for the post-processed 2D CNN predictions produced by the individually trained model of each cross-validation fold, using 5,000 voxels picked randomly from each tomogram of the respective test set of cross-validation fold. Precision and recall (Supplementary Note 1 and Supplementary Fig. 4) were computed at threshold values varying from 0 to 1 on the picked voxels to compute the area under the precision–recall curves (AURPC) for each tomogram.

## Hyperparameter tuning of DeePiCt

The effect of user-defined hyperparameters of the 3D CNN (IF, D, ED, DD, and BN) were evaluated using the cross-validation and performance metrics described above. Starting with the default hyperparameter combination (Supplementary Note 1), we tested first for different values of D, while keeping the rest fixed. Then, we fixed D for which best performance was achieved and repeated the same process for IF. We continued with each of the remaining parameters, BN, ED and DD.

In all cases, we combined the 3D CNN segmentation of the target structure with an appropriate 'region mask' to eliminate false positives (Supplementary Note 1 and Supplementary Table 1). For ribosome, FAS and membrane, both 3D CNN (for target structure prediction) and 2D CNN (for 'region mask' prediction) in DeePiCt employed a threefold cross-validation in the VPP dataset as described above (Extended Data Fig. 6a–c).

## Computational setup

All experiments for the 3D CNN of the DeePiCt pipeline and Deep-Finder were performed using NVIDIA 2080 Ti GPU, Cuda 10.0, Python 3 and Pytorch 1.3.1. For the 2D CNN, training was conducted using an NVIDIA 2080 Ti GPU and an NVIDIA V100S GPU used for performance evaluation, using CUDA 10.0, Python 3 and Keras 2.3.1 with a tensorflow 2.0.0 backend. Detailed lists of parameters used for the 2D and 3D CNN are available alongside the DeePiCt source code (Code Availability).

## Reporting summary

Further information on research design is available in the Nature Portfolio Reporting Summary linked to this article.

## Data availability

Raw tilt-series, tomograms, ground truth coordinates, and segmentations are available via EMPIAR accession codes EMPIAR-10988 (*S. pombe*) and EMPIAR-10989, EMD-16136 (RPE-1). Subtomogram averages for *S. pombe* VPP and defocus ground truth annotations are available from the EMDB: VPP ground truth, EMD-14404, EMD-14405, EMD-14406, EMD-14408, EMD-14409, EMD-14410, EMD-14411; defocus ground truth, EMD-14412, EMD-14413, EMD-14415, EMD-14417, EMD-14418, EMD-14419, EMD-14420; defocus DeePiCt predicted, EMD-14422, EMD-14423, EMD-14424, EMD-14425, EMD-14426. Structural comparisons were performed with *S. cerevisiae* FAS (PDB accession 2UV8 (ref. [31])), *P. pastoris* FAS (EMDB accession EMD-12139 (ref. [32])), eEF3 from *S. cerevisiae* (EMDB accession EMD-12062 (ref. [37])), the *S. cerevisiae* ribosome (EMDB accession EMD-1667 (ref. [38])) with the rRNA expansion segment ES27L (PDB accession 3IZD (ref. [39])), the nuclear export factor Arx1 bound to the 60S large ribosomal subunit *S. cerevisiae* (EMDB accession EMD-2169 (ref. [41])), the human Ebp1 (EMDB accession EMD-10608 (ref. [42])), *S. cerevisiae* ribosomes derived from extracted ER (EMDB accession EMD-3764 (ref. [44])), and the ER-bound HeLa ribosomes (EMDB accession EMD-8056 (ref. [45])). The large subunit (LSU, 60S) of a published *S. cerevisiae* 80S ribosome map (EMDB accession EMD-3228 (ref. [61])) and the *S. cerevisiae* FAS map (EMDB accession EMD-1623 (ref. [33])) were used as references for template matching. The HeLa cell dataset is available via EMDB accession EMD-11992 (ref. [45]).

## Code availability

The DeePiCt code is assembled as two Snakemake pipelines (2D CNN and 3D CNN); the 3D CNN is implemented in the Pytorch framework while the 2D CNN is implemented in Keras, both in Python 3. The code, trained models, link to the Google Colab notebook, including custom scripts for the extraction of particle coordinates from filaments are available in the GitHub repository https://github.com/ZauggGroup/DeePiCt.

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

## Acknowledgements

We thank J. Zagoriy and W. Hagen for technical support, EMBL IT and especially T. Hoffmann for computational support, C. Haering for kindly providing the *S. pombe* and J. B. Woodruff for the *C. elegans*. I.d.T.-T. and M.T.-N. were supported by a fellowship from the EMBL Interdisciplinary Postdoctoral Program (EI3POD) under Marie Skłodowska-Curie Actions COFUND. D.W.C.C. was supported by the Boehringer Ingelheim Fonds PhD fellowship and the Croucher Scholarship for Doctoral Study. J.M. and J.B.Z. acknowledge funding from the EMBL. J.M. acknowledges funding from the European Research Council starting grant (3DCellPhase⁻760067).

## Author contributions

I.d.T.-T., J.M. and J.B.Z. conceived the study. I.d.T.-T. implemented the 3D CNN code, co-produced ground truth annotations, and carried out numerical experiments to evaluate performance. S.K.G. prepared *S. pombe* cryo-FIB lamellae and acquired cryo-electron tomograms, co-produced ground truth annotations, and carried out subtomogram averaging and structural analysis. A.M. implemented the 2D CNN and spectrum equalization filter, produced ground truth segmentations, and conducted performance analysis of the 2D CNN. F.S. carried out training and prediction with 3D CNN networks, implemented code, and created Colab Notebooks for prediction. C.E.Z. generated data and provided NPC annotations under the supervision of M.B. M.T.-N. generated RPE-1 and MEF 3T3 data and together with D.W.C.C. produced ground truth and trained networks for actin segmentation. F.T. generated data, ground truth, trained networks for microtubule segmentation, and generated the subtomogram average. C.P. and A.K. provided valuable input for the design and generation of the CNNs. A.D.-M. provided input and supervision for the actin segmentation. I.d.T.-T., S.K.G., A.M., J.M., and J.Z. wrote the manuscript with input from all authors.

## Competing interests

The authors declare no competing interests.

## Additional information

**Extended data** is available for this paper at https://doi.org/10.1038/s41592-022-01746-2.

**Correspondence and requests for materials** should be addressed to Julia Mahamid or Judith B. Zaugg.

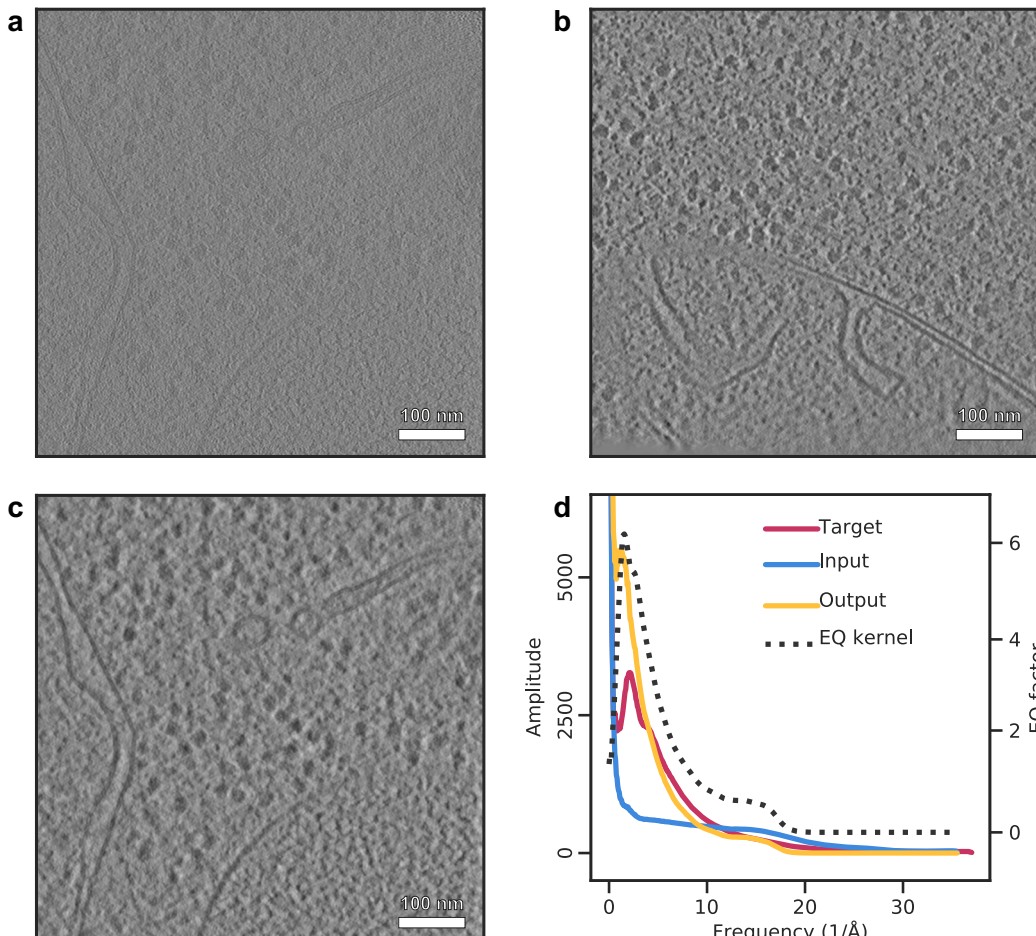

**Extended Data Fig. 1 | Pre−processing of input data by spectrum equalization. a**, Input defocus tomogram with low SNR. **b**, Target VPP tomogram (one example from $n = 10$ tomograms) with high SNR. **c**, Output showing the tomogram in **a** after spectrum matching. **d**, Amplitude spectra of target, input and output tomograms, as well as per-frequency (1/Å) scaling factors of the rotational equalization (EQ) kernel (y-axis cropped).

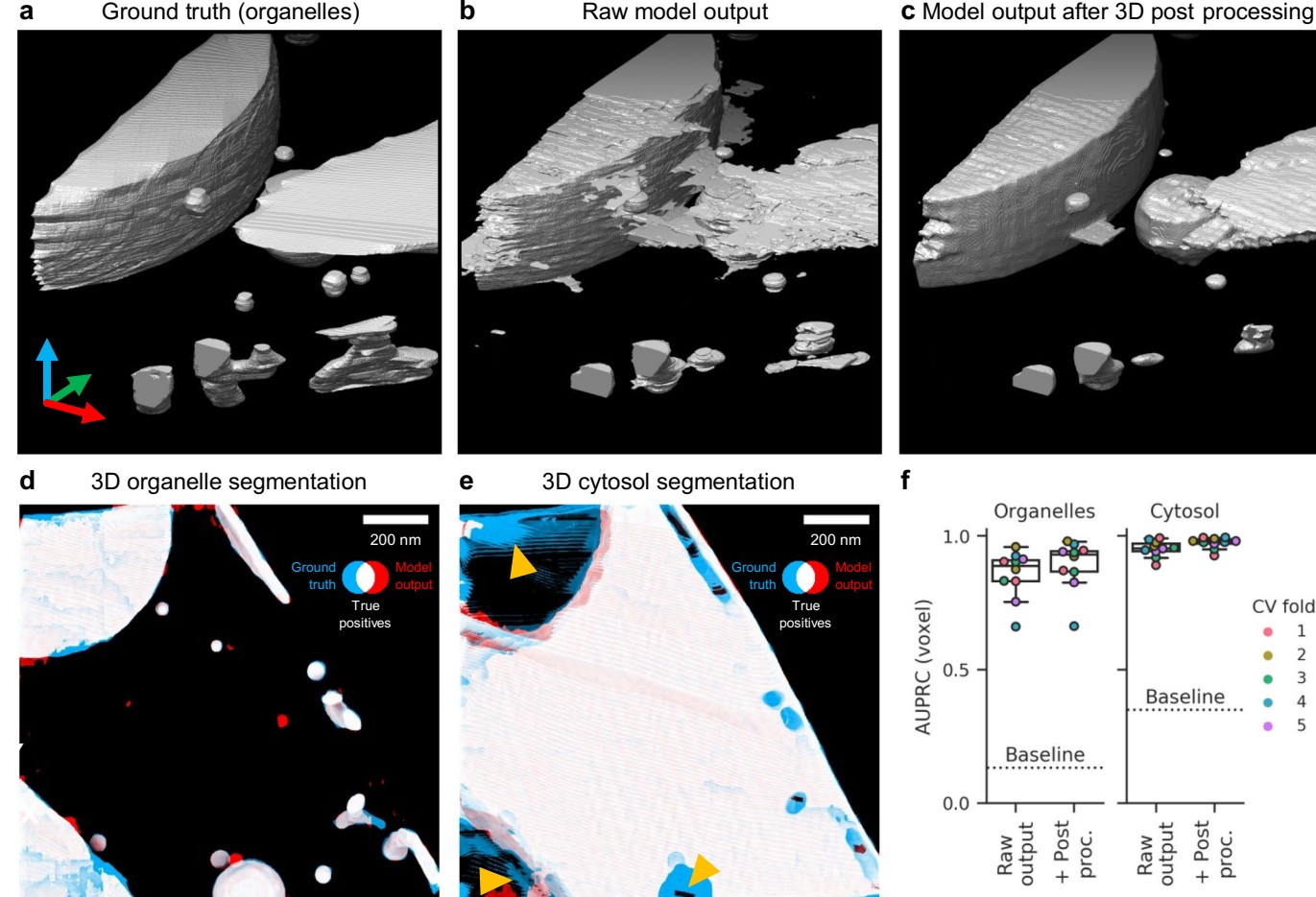

**Extended Data Fig. 2 | 2D CNN organelle and cytosol segmentations.**
**a-c**, Isosurface view of organelle ground truth manual annotations (**a**), model output prior to (**b**) and after 3D post-processing (**c**). **d-f**, Organelle (**d**) and cytosol (**e**) predictions of the tomogram in Fig. 1b. The 2D CNN's segmentations (red) are overlaid with the manually created ground truth (blue), with overlapping regions (that is, true positives) in white. Ground truth annotations in the top and bottom left corners in **e** (yellow arrowheads) are artifacts resulting from cross-slice interpolation in the manual annotation process. **f**, 3D post-processing improves performance of the 2D CNN as exemplified in (**a-c**). Plots show results for $n = 2$ tomograms over 5 independent experiments. Boxplots middle line marks the median and the edges indicate the 25th and 75th percentiles; whiskers encompass all data that are not considered outliers (calculated by the Seaborn boxplot function).

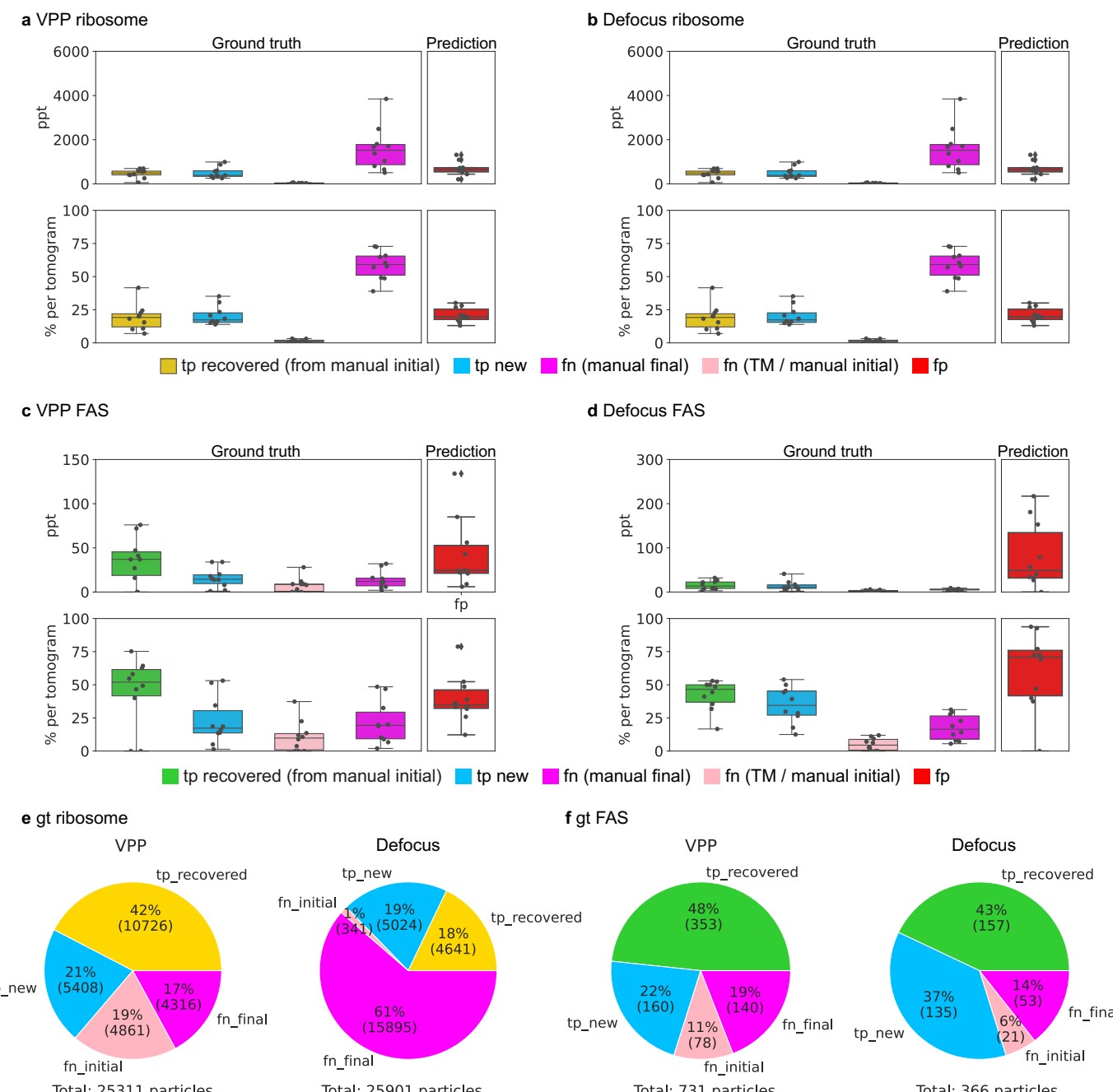

**Extended Data Fig. 3 | Details of the construction of ground truth for ribosome and FAS in VPP and defocus tomograms. a,b**, Contributions of the steps applied during the construction of ribosome ground truth in the VPP (**a**) and defocus (**b**) datasets (*n = 10* tomograms each). Full data provided in Supplementary Table 2. Upper subplots show the absolute number of particles per tomogram (p.p.t.): true positives (tp) from the initial annotations that were recovered in step 2 (tp recovered; yellow), newly identified true positives in step 2 (tp new; blue), particles that were in the initial annotation but that were not recovered in step 2 (false negatives from initial annotation; fn initial; pink), unidentified particles from any of the two first steps, which were manually picked in step 3 (fn final; fuchsia), and total false positives (fp; red) in step 2. Lower subplots show the associated relative contribution, as a percentage of the ground truth per tomogram (ground truth panel) or as percentage of the predictions per tomogram in step 2 (prediction panel). **c,d**, Equivalent plots for FAS ground truth construction, where the only difference is that the recovered true positives originate from an initial manual annotation (green), as opposed to ribosomes where we additionally used TM. Full data provided in Supplementary Table 3. Boxplots middle line marks the median and the edges indicate the 25th and 75th percentiles; whiskers encompass all data that are not considered outliers (calculated by the Seaborn boxplot function). **e,f**, Summary of the relative contributions in the ribosome (**e**) and FAS (**f**) ground truth (gt) across the 10 VPP and 10 defocus tomograms.

**a**

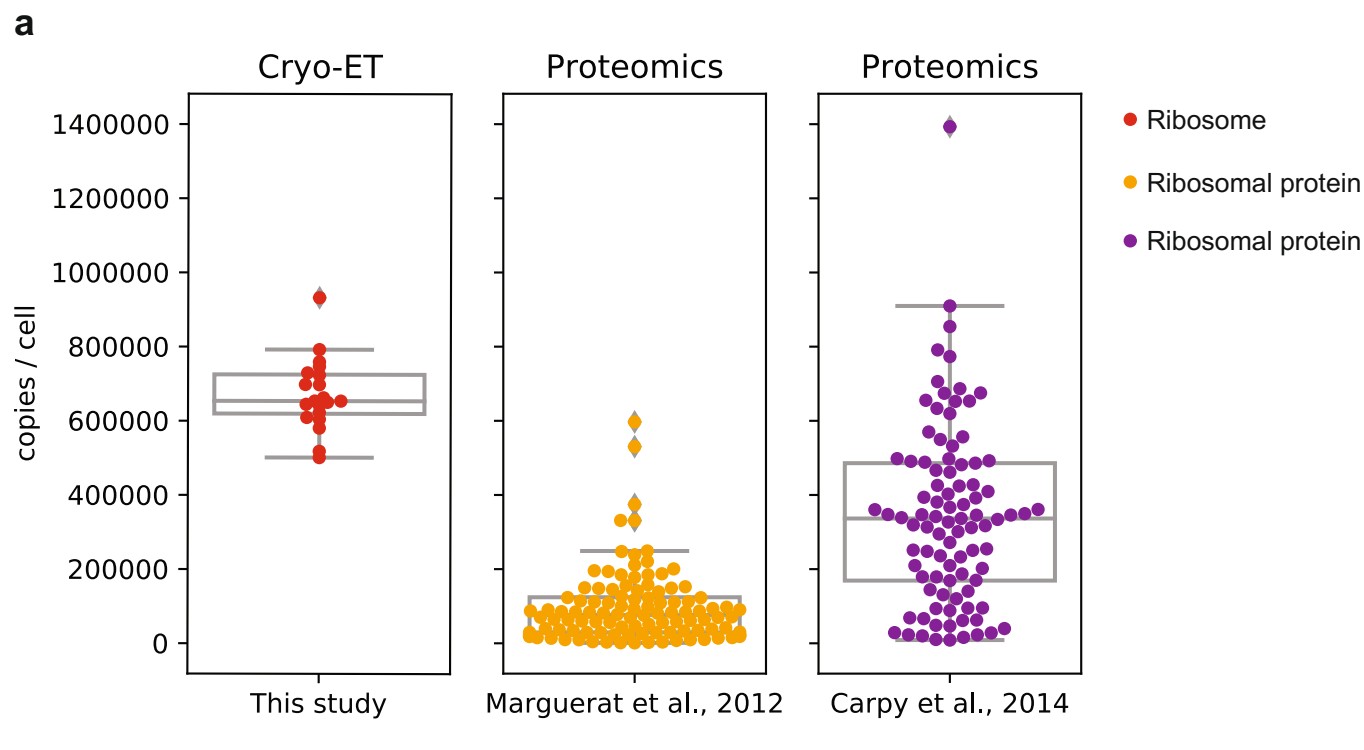

**b**

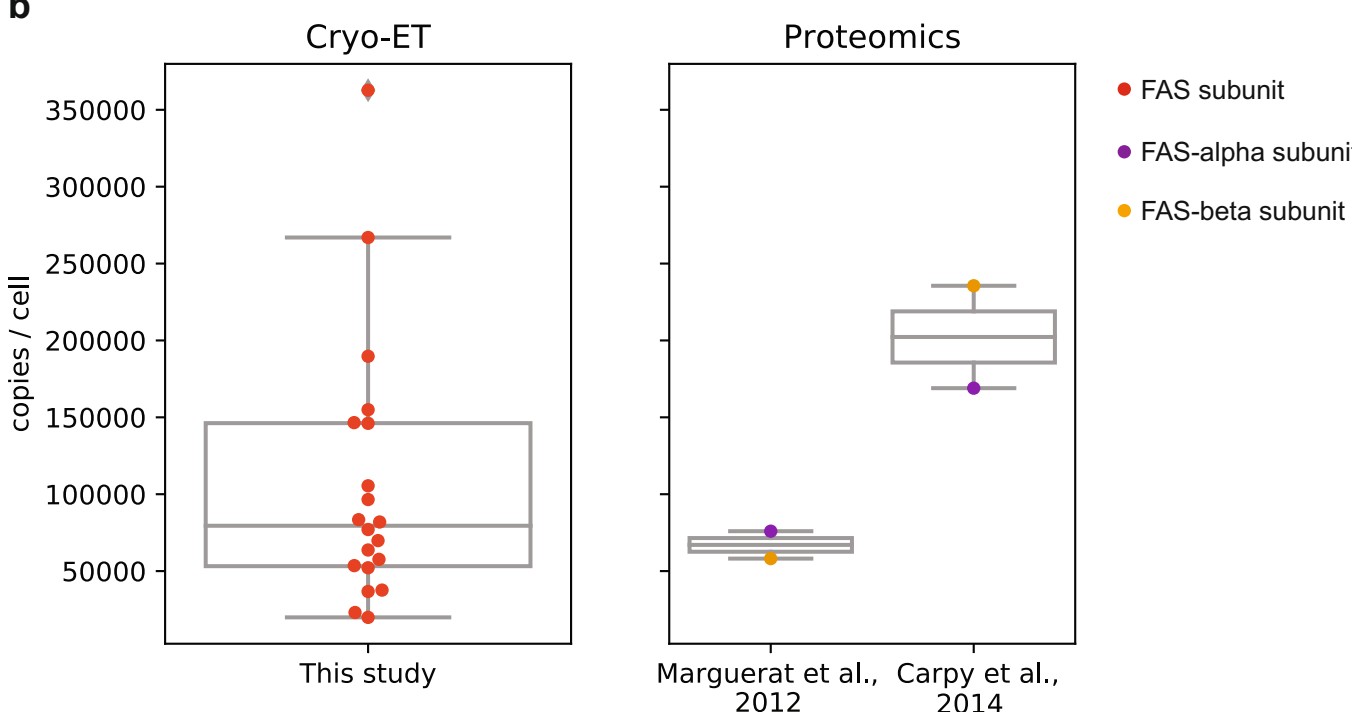

**Extended Data Fig. 4 | *S. pombe* ribosome and FAS complex abundances in cryo-electron tomograms in comparison to proteomics analysis.** In this study (cryo-ET, left), fully assembled ribosomes (**a**) and FAS (**b**) were iteratively annotated in 20 ground truth cryo-electron tomograms and cytosolic concentrations calculated using the cytosol segmentations. Copy numbers per cell were calculated for an *S. pombe* cell with the assumption of 30 % cytosolic volume[62] of a total of 150 μm³ average cell volume[63,64]. **a**, With cryo-ET, an average of 671,303 ± 96,708 ribosomes/cell were annotated. Ribosome counts from individual tomograms are represented as individual points (red). Proteomics data of individual ribosomal proteins were derived from Marguerat et al.[28] (yellow) and Carpy et al.[27] (purple) and resulted in average of 97,795 ± 99,298

and 343,511 ± 244,226 ribosomes/cell, respectively. **b**, Each measurement displayed in the plot corresponds to 6 times fully assembled FAS counts per tomogram (the complex is constituted by six alpha and beta subunits). With cryo-ET, an average of 106,282 ± 86,247 FAS subunits/cell were observed. Proteomics data of individual FAS proteins (subunit alpha in purple, beta in yellow) were derived from Marguerat et al.[28] and Carpy et al.[27] and resulted in average of 67,035 ± 12,600 and 202,263 ± 47,086 FAS subunits/cell, respectively. Boxplots middle line marks the median and the edges indicate the 25th and 75th percentiles; whiskers encompass all data that are not considered outliers (calculated by the Seaborn boxplot function).

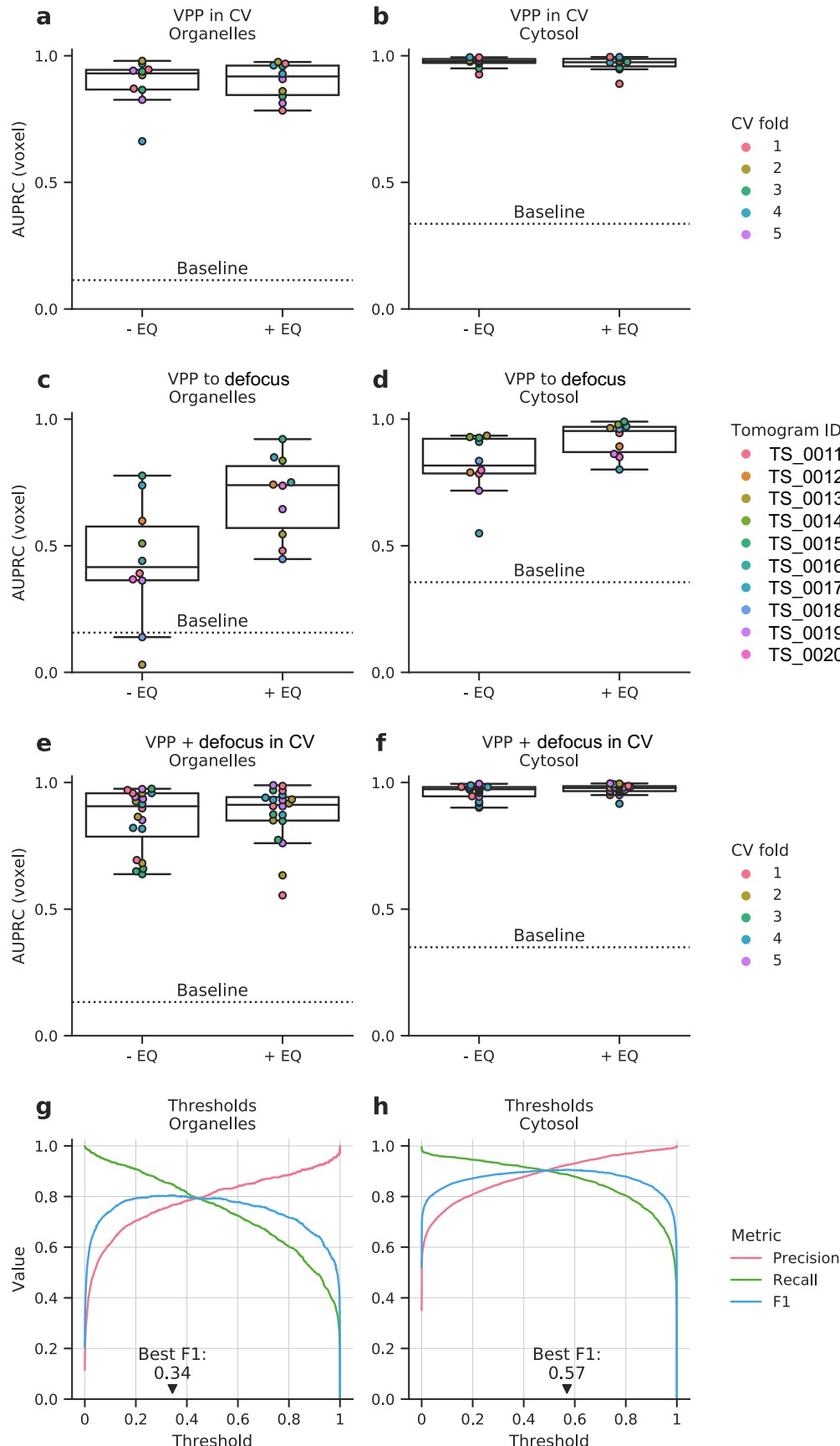

**Extended Data Fig. 5 | See next page for caption.**

**Extended Data Fig. 5 | Effects of spectrum equalization on cross-validation and performance evaluation across domains of the 2D CNN. a-f**, Performance was evaluated for 2D networks in three different scenarios (training on VPP in cross-validation (CV), cross-domain generalization from VPP to defocus, training on VPP and defocus in cross-validation), for either organelles or cytoplasm, without (-EQ) and with (+EQ) spectrum equalization. Areas under the precision-recall curves (AUPRCs) were computed by randomly picking 1000 voxels from each tomogram in test set (for the two cross-validation scenarios) or from the defocus tomograms (for the cross-domain tasks), while restricting evaluation to z-slices with any positive label. AUPRCs were computed after 3D post-processing. The baselines are defined as the fraction of positive labels within those z-slices and averaged across test tomograms. Boxplots middle line marks the median and the edges indicate the 25th and 75th percentiles; whiskers encompass all data that are not considered outliers (calculated by the Seaborn boxplot function). **g,h**, Precision, recall and F1 score of organelle (**g**) and cytosol (**h**) segmentation depending on probability threshold used. Scores were computed on the cross-validation results for the spectrum-equalized VPP tomograms.

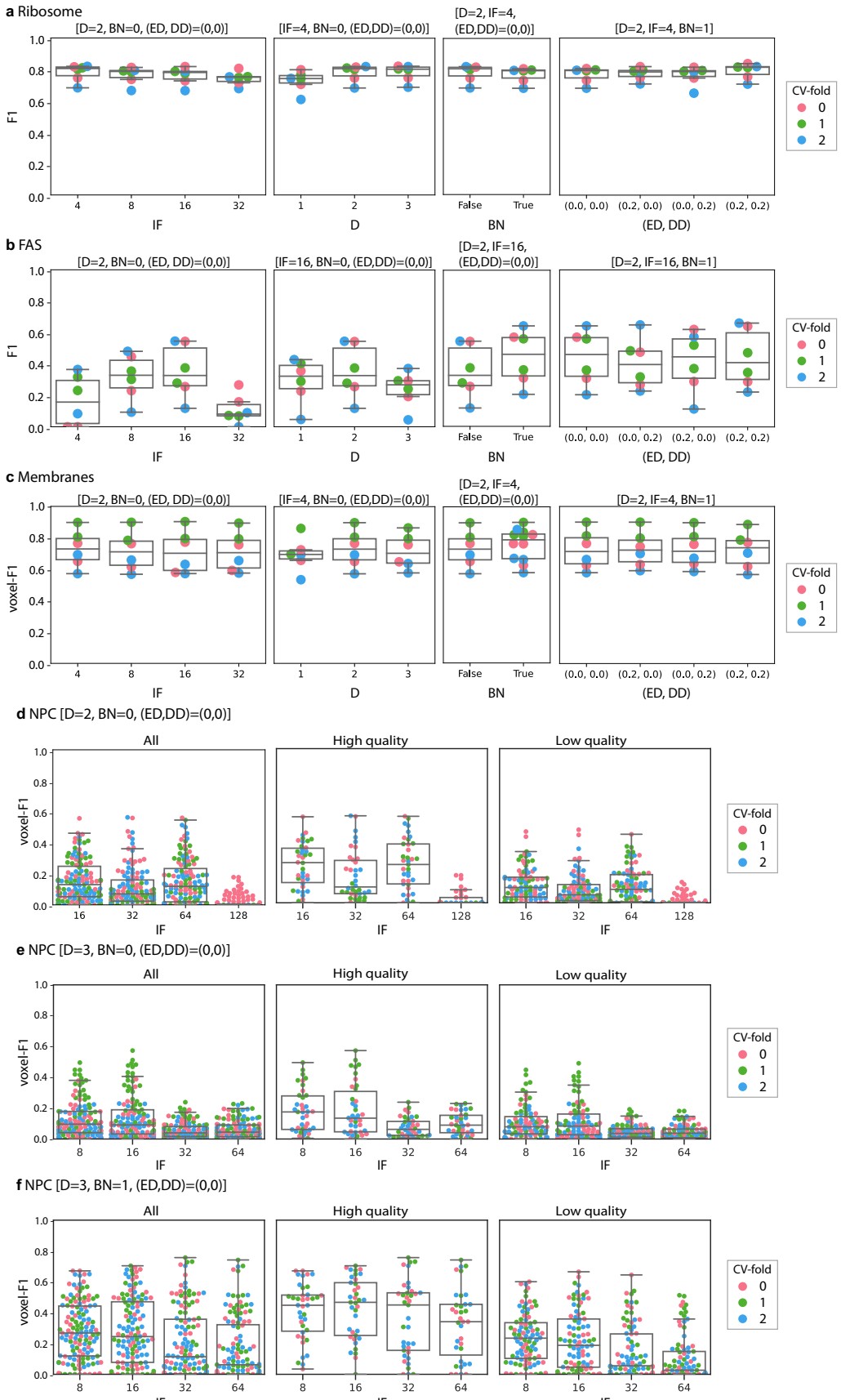

**Extended Data Fig. 6 | See next page for caption.**

**Extended Data Fig. 6 | Hyperparameter tuning and performance analysis of 3D CNN for ribosome, FAS, membranes and NPCs.** Hyperparameters: D: depth, IF: number of initial features, BN: batch normalization layers, DD: decoder dropout, ED: encoder dropout. **a-c**, Cross validation (CV) analysis for the performance for ribosomes, FAS and membranes, respectively ($n = 2$ tomograms over 3 independent experiments). In these experiments, we follow an incremental search: from left to right, the starting subplot corresponds to the default configuration of the 3D CNN, D = 2, BN = 0, and (ED, DD) = (0,0), except for variable IF. The best value of IF is fixed in the subsequent subplot to the right, where D is varied over 1, 2, and 3. The next subplot to the right compares the effect of not applying and applying the batch normalization layer (BN = 0 and BN = 1, respectively), under the best IF and D from previous subplots. Finally, the right most subplot varies the encoder- and decoder-dropout parameters (ED, DD), with fixed best IF, D, and BN from previous subplots. **d-f**, Results of hyperparameter exploration for NPC in all defocus* tomograms (left, $n = 127$ tomograms), in the subset of high-quality defocus* tomograms (middle), and in the subset of lower quality defocus* tomograms (right) ($n = 42$, $n = 13$ and $n = 39$ tomograms over the same 3 independent experiments, respectively). **d**, Shows the variations in IF, when D = 2, BN = 0, and (ED, DD) = (0,0); **e**, Variations in IF, when D = 3, BN = 0, and (ED, DD) = (0,0); **f**, Variations in IF, when D = 3, BN = 1, and (ED, DD) = (0,0). The summary of best hyperparameter combinations is provided in Supplementary Table 1. Boxplots middle line marks the median and the edges indicate the 25th and 75th percentiles; whiskers encompass all data that are not considered outliers (calculated by the Seaborn boxplot function).

Spectrum equalization preprocessing (EQ) effect

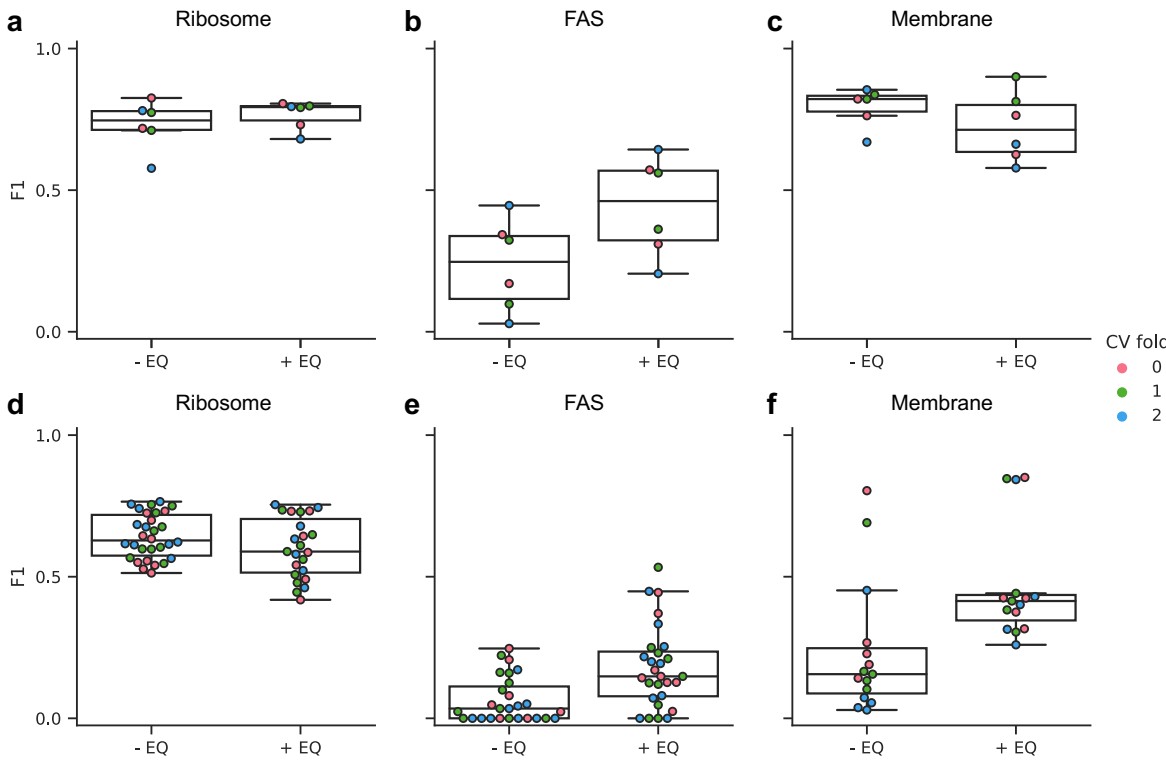

Cytosol masking effect

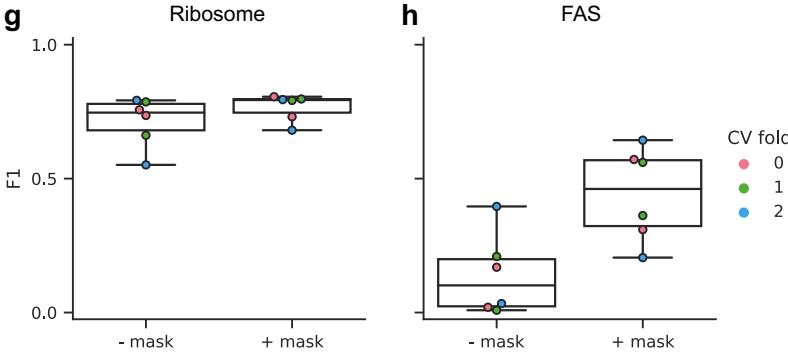

**Extended Data Fig. 7 | Effects of spectrum equalization and cytosol masking on DeePiCt performance. a-c**, Performance of the DeePiCt pipeline within the VPP domain for ribosomes, FAS and membranes, without (-EQ) and with (+EQ) spectrum equalization pre-processing ($n = 2$ tomograms, over 3 independent experiments). The F1 score distribution shows that although for ribosomes and membranes the spectrum equalization does not have a clear positive effect, for FAS it brings better results with a shift in median F1 score from 0.25 to 0.46. **d-f**, The same networks were used to test performance of the DeePiCt pipeline across the VPP and defocus domains (training on VPP and testing in defocus) for ribosomes ($n = 10$ tomograms), FAS ($n = 10$) and membranes ($n = 5$) without (-EQ) and with (+EQ) spectrum equalization pre-processing. The F1 score distribution shows that while for ribosomes the spectrum equalization does not bring benefits, for FAS and membrane prediction across domains, its use brings a positive median F1 shift from 0.03 to 0.15 and from 0.16 to 0.41, respectively. **g-h**, Effects of employing a segmentation of cytosol as a *region mask* (+ mask) or not (- mask) during DeePiCt's post-processing, for the localization of ribosome and FAS. The plots show the results for $n = 2$ tomograms over 3 independent experiments (3-fold cross-validation) in VPP data, where the cytosol segmentation was obtained by a 2D CNN. **g**, Ribosome localization does not show a significant improvement in F1 score when the cytosol masking is used. **h**, FAS results show a stronger difference in F1 score, with a shift of median F1 from 0.10 to 0.46. Boxplots middle line marks the median and the edges indicate the 25th and 75th percentiles; whiskers encompass all data that are not considered outliers (calculated by the Seaborn boxplot function).

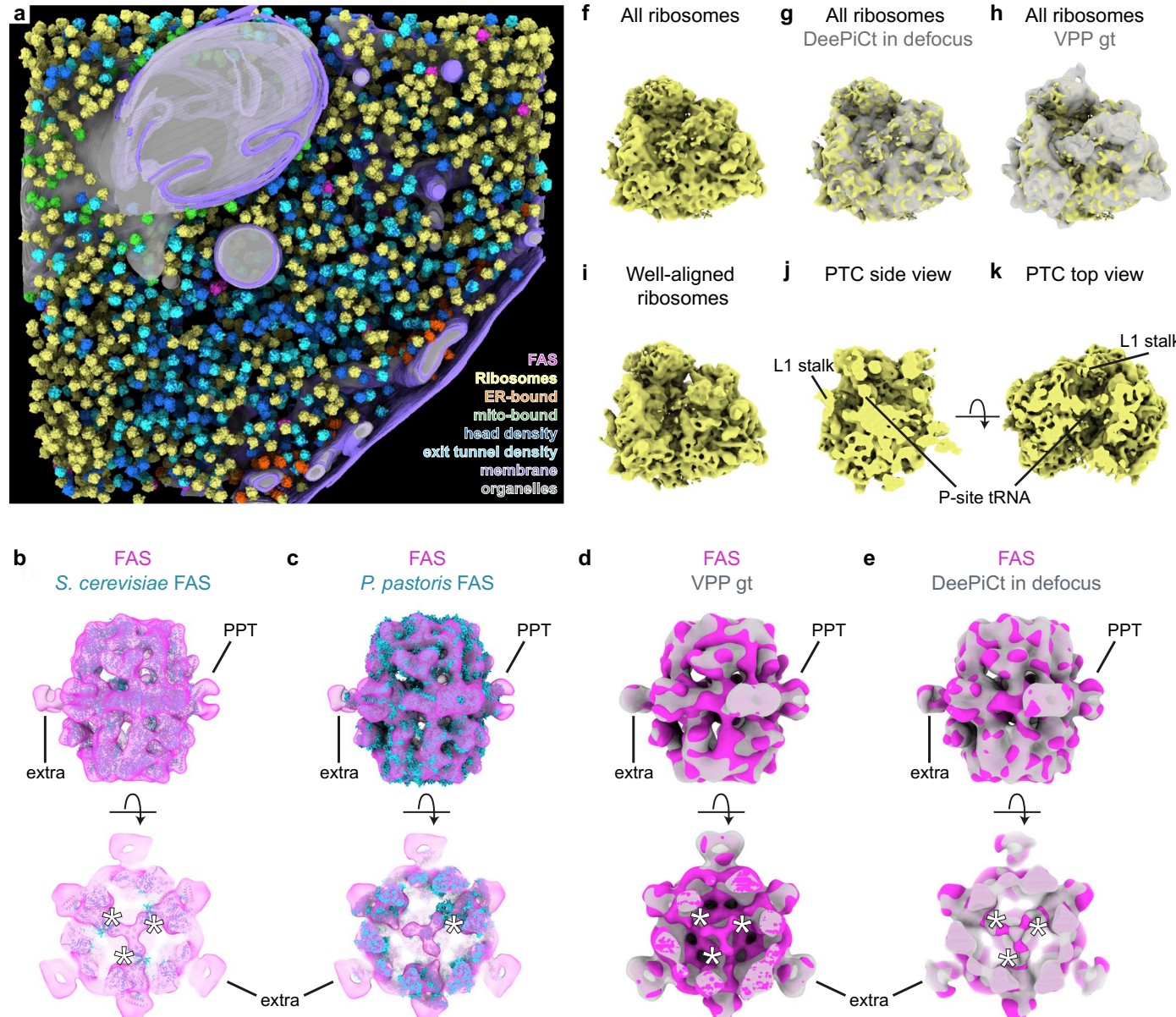

**Extended Data Fig. 8 | Ground truth annotations and 3D refinement of ribosomes and FAS in defocus tomograms. a**, Complete ground truth annotation of ribosomes (yellow, orange, green, dark and light blue represent sub-populations described in (**f**) and in Extended Data Fig. 9), FAS (pink), membranes (purple) and organelles (gray) in the tomogram depicted in Fig. 4. Ribosome subclasses in the context of specific organelles were identified by a maximal distance of 25 nm to mitochondria (mito-bound, green) or ER (ER-bound, orange). Focused classification on densities close to the head of the 40 S small ribosomal subunit or in proximity to the exit tunnel recovered ribosome subsets with head (dark blue) and exit tunnel density (cyan). **b-e**, FAS subtomogram averages (pink) fitted with the *S. cerevisiae* X-ray structure (**b**, cyan) and the single particle cryo-EM density of *P. pastoris* (**c**, cyan) including the

PPT domains. An extra density cannot be assigned. The cross section views close to the alpha-wheel reveal three densities fitting three and one ACPs of *S. cerevisiae* and *P. pastoris*, respectively (white asterisks, bottom). **d**, Overlay with FAS from VPP ground truth (gt) tomograms (gray) shows overall similarity, but the ACPs could not be resolved (white asterisks, bottom). **e**, FAS predicted by DeePiCt (gray) matches the defocus ground truth average with three ACP densities close to the alpha-wheel. **f-h,** Subtomogram average of all ribosomes (yellow) matches the DeePiCt-predicted (**g**, gray) and VPP-derived (**h**, gray) ribosome densities. **i-k**, Well-aligned ribosomes (yellow) detected by hierarchical 3D classification in RELION and refined in M (Supplementary Fig. 7). **j-k**. Slicing through the average depicted in (**i**) at different axes reveals the PTC with a P-site tRNA and the L1 stalk facing the E-site.

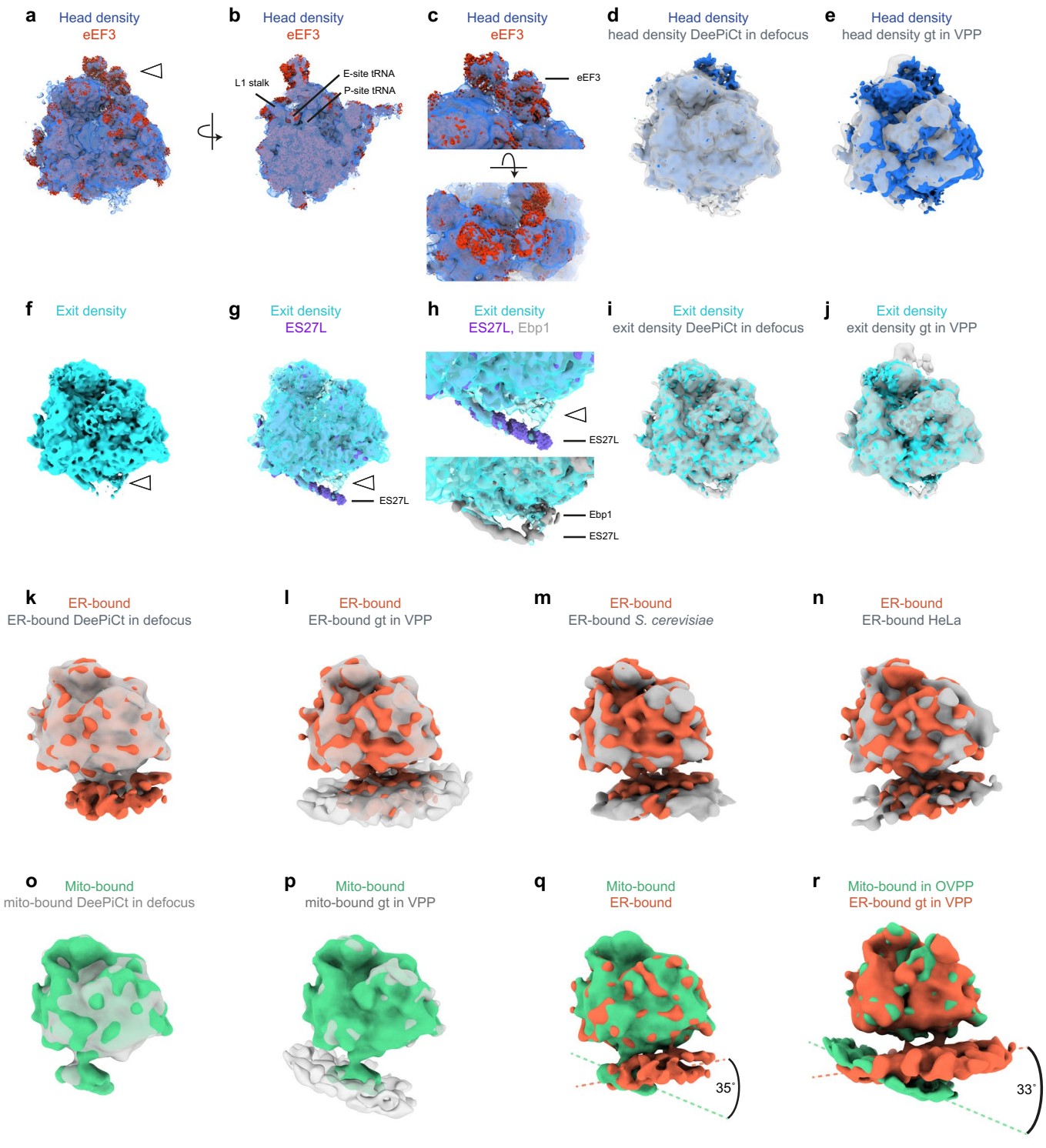

**Extended Data Fig. 9 | Comparison of ribosome subpopulations in defocus ground truth (gt). a-c**, Subtomogram average of ribosomes (blue) with electron density in proximity to the head of the 40 S small subunit (white arrowhead) which fits the ribosome-bound eEF3[37] (CC 0.8972, red). **b**, Cross section of (**a**) showing the PTC with P- and E-site tRNAs and the L1 stalk facing inwards. **c**, Zoom into the additional head density fitting eEF3. **d, e**, Overlay with the same ribosome class from DeePiCt predictions in defocus (**d**, gray) and from VPP gt data (**e**, gray). **f**, Ribosome subset (cyan) with electron density in proximity to the exit tunnel (white arrowhead). **g**, Overlay with the *S. cerevisiae* ribosome[38] (CC 0.9651, purple) and a specific conformation of the expansion segment ES27L[39] (CC 0.7806, purple). **h**, Zoom into the exit tunnel area in (**g**) highlights the additional density connected to ES27L that colocalizes with for example the human Ebp1[42] (gray, lower panel, CC 0.8495, EMD-10608). **i,j**, Overlay with

the same ribosome class from DeePiCt predictions in defocus (**i**, gray) and from VPP gt data (**j**, gray). **k-n**, Overlay of ribosomes within 25 nm distance to the ER (orange) with ER-bound ribosomes from DeePiCt predictions in defocus (**k**, gray), from VPP gt data (**l**, gray), with the published density of *S. cerevisiae* ribosomes derived from extracted ER[44] (**m**, gray, CC 0.8972, EMD-3764), and the ER-bound HeLa ribosome[45] (**n**, gray, CC 0.9105, EMD-8056). All densities show the same interface between ribosome and ER membrane. **o,p**, Overlay of ribosomes within 25 nm distance to mitochondria (green) with mito-bound ribosomes from DeePiCt predictions in defocus (**o**, gray), and VPP gt data (**p**, gray). All densities show the same interface between ribosome and mitochondria membrane. **q,r**, Comparisons between ER-bound (orange) and mitochondria-bound (mito-bound, green) subtomogram averages in defocus gt (**q**) and VPP gt (**r**) data reveal an angular offset of the membrane interfaces of 35° and 33°, respectively.

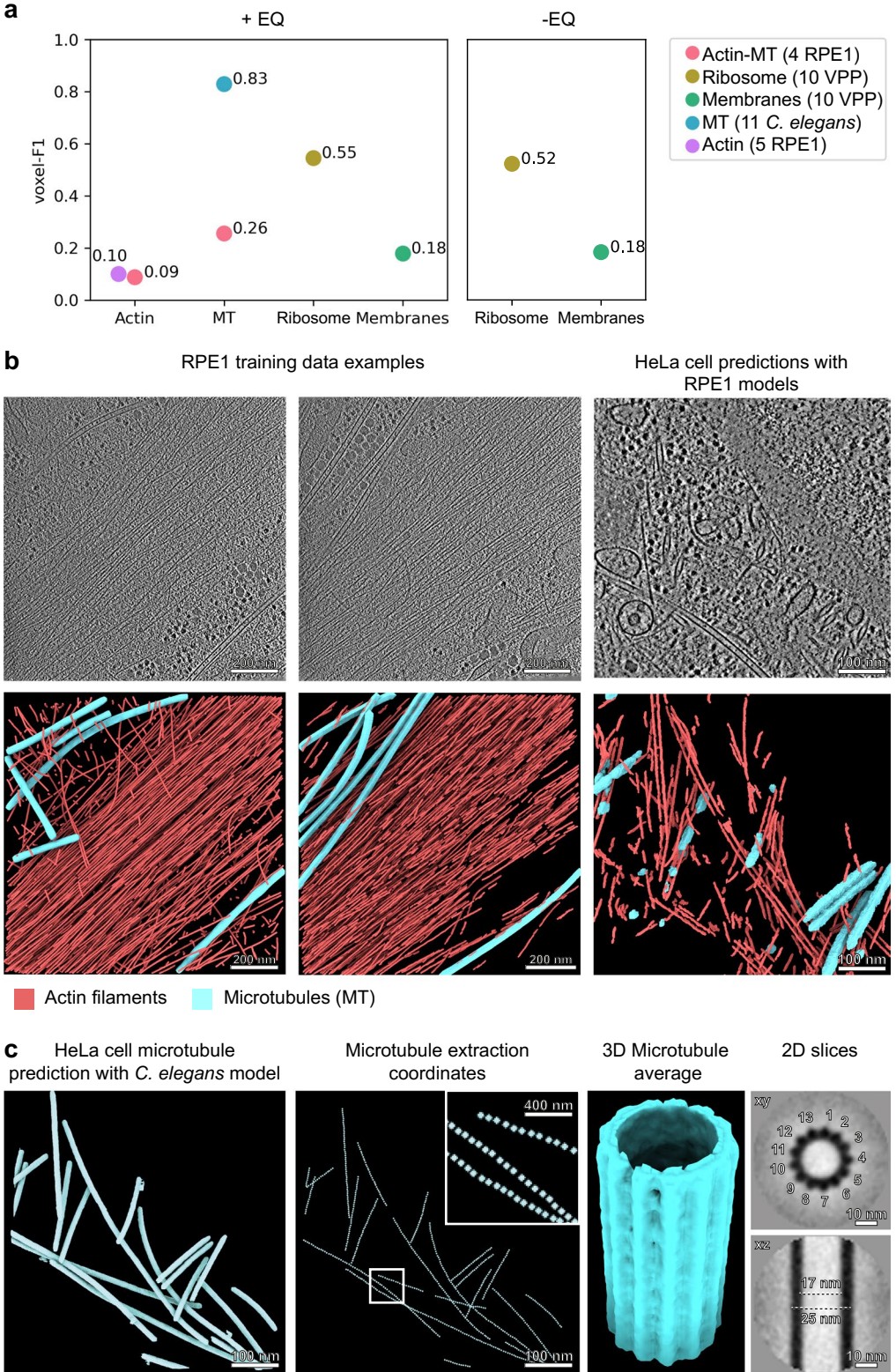

**Extended Data Fig. 10 | See next page for caption.**

**Extended Data Fig. 10 | Domain generalization of different cellular structures and species. a**, Performance results of five DeePiCt networks trained on a variety of cell types for the segmentation of different cellular structures used for predictions in a HeLa cell dataset[45]. Left: plot of performance scores when employing the spectrum equalization (+EQ) filter in both the training and the HeLa cell data. The results correspond to segmentations of actin filaments and microtubules (MT) (trained on 4 RPE1 tomograms; pink), ribosomes (yellow-green) and membranes (aquamarine) trained on 10 *S. pombe* VPP tomograms, MT (trained on 11 *C. elegans* tomograms; blue) and actin filaments (trained on 2 RPE1 and 3 MEF 3T3 tomograms with actin filaments at different orientations; light purple). Right: performance for ribosomes and membranes without employing the equalization filter (-EQ) in the HeLa cell dataset. **b**, RPE1 training data examples (left and middle, tomograms on top, ground truth annotations at the bottom) and predictions (right) of the actin filament and MT models in the HeLa cell dataset. The training examples show 2 of the RPE1 tomograms[50] used for this model. The performance of actin and microtubule predictions in the HeLa cell dataset is poor (voxel-F1 = 0.09 for actin and 0.26 for MT; **a** left) despite the high-quality training data (left), likely due to the high unidirectional orientation of the filaments in the training data. **c**, Predictions in the HeLa cell dataset (left) of MT model trained with *C. elegans* tomograms (voxel-F1 = 0.83). Coordinates (middle) were extracted from the binary prediction map (left) using custom scripts (Code Availability). The inset shows a zoom into the boxed area. A 3D microtubule average (right, cyan) at 39 Å resolution was generated using Dynamo[24]. 2D slices through the volume show the xy plane (top) with the microtubule protofilaments labeled, and the xz plane (bottom) with the 25 nm microtubule and 17 nm lumen diameter indicated.

# nature research

# Reporting Summary

Nature Research wishes to improve the reproducibility of the work that we publish. This form provides structure for consistency and transparency in reporting. For further information on Nature Research policies, see our Editorial Policies and the Editorial Policy Checklist.

## Statistics

For all statistical analyses, confirm that the following items are present in the figure legend, table legend, main text, or Methods section.

| n/a | Confirmed | |
|---|---|---|
| ☐ | ☒ | The exact sample size (*n*) for each experimental group/condition, given as a discrete number and unit of measurement |
| ☐ | ☒ | A statement on whether measurements were taken from distinct samples or whether the same sample was measured repeatedly |
| ☐ | ☒ | The statistical test(s) used AND whether they are one- or two-sided *Only common tests should be described solely by name; describe more complex techniques in the Methods section.* |
| ☐ | ☒ | A description of all covariates tested |
| ☒ | ☐ | A description of any assumptions or corrections, such as tests of normality and adjustment for multiple comparisons |
| ☐ | ☒ | A full description of the statistical parameters including central tendency (e.g. means) or other basic estimates (e.g. regression coefficient) AND variation (e.g. standard deviation) or associated estimates of uncertainty (e.g. confidence intervals) |
| ☐ | ☒ | For null hypothesis testing, the test statistic (e.g. *F*, *t*, *r*) with confidence intervals, effect sizes, degrees of freedom and *P* value noted *Give P values as exact values whenever suitable.* |
| ☒ | ☐ | For Bayesian analysis, information on the choice of priors and Markov chain Monte Carlo settings |
| ☒ | ☐ | For hierarchical and complex designs, identification of the appropriate level for tests and full reporting of outcomes |
| ☒ | ☐ | Estimates of effect sizes (e.g. Cohen's *d*, Pearson's *r*), indicating how they were calculated |

*Our web collection on statistics for biologists contains articles on many of the points above.*

## Software and code

Policy information about availability of computer code

| Data collection | SerialEM 3.7 or 3.8: Mastronarde, D.N. 2005. Automated electron microscope tomography using robust prediction of specimen movements. J. Struct. Biol. 152:36-51 (doi:10.1016/j.jsb.2005.07.007) |
|---|---|
| Data analysis | The DeePiCt code is assembled as two Snakemake pipelines (2D CNN and 3D CNN); the 3D CNN is implemented in the Pytorch framework while the 2D CNN is implemented in Keras, both in Python 3. The code, trained models, and link to the Google Colab notebook are available in the github repository https://github.com/ZauggGroup/DeePiCt. Cryo-electron tomograms were reconstructed with IMOD (version 4.9.4, doi: 10.1006/jsbi.1996.0013). Structural data analysis was performed with Warp (version 1.0.9, doi: 10.1038/s41592-019-0580-y), M (version 1.0.9, doi: 10.1038/s41592-020-01054-7), Relion (version 3.0.7, doi:10.7554/eLife.42166), Dynamo (version 1.1.520, doi: 10.1107/S2059798317003369), and TOM toolbox release-2008 (doi: 10.1016/j.jsb.2004.10.006) implemented in MATLAB 2016b & 2019b (https://www.mathworks.com). Subtomogram averages were visualized with UCSF Chimera (version 1.16.0, doi: 10.1002/jcc.20084) and UCSF ChimeraX (version 1.4.0, doi: 10.1002/pro.3235). |

For manuscripts utilizing custom algorithms or software that are central to the research but not yet described in published literature, software must be made available to editors and reviewers. We strongly encourage code deposition in a community repository (e.g. GitHub). See the Nature Research guidelines for submitting code & software for further information.

## Data

Policy information about availability of data

All manuscripts must include a data availability statement. This statement should provide the following information, where applicable:
- Accession codes, unique identifiers, or web links for publicly available datasets
- A list of figures that have associated raw data
- A description of any restrictions on data availability

Raw tilt series, tomograms, ground truth coordinates and segmentations are available via EMPIAR accession codes EMPIAR-10988 (S. pombe) and EMPIAR-10989, EMD-16136 (RPE1). Subtomogram averages for S. pombe VPP and defocus ground truth annotations are available on EMDB (VPP ground truth: EMD-14404, EMD-14405, EMD-14406, EMD-14408, EMD-14409, EMD-14410, EMD-14411; defocus ground truth: EMD-14412, EMD-14413, EMD-14415, EMD-14417, EMD-14418, EMD-14419, EMD-14420; defocus DeePiCt predicted: EMD-14422, EMD-14423, EMD-14424, EMD-14425, EMD-14426).

Structural comparisons were performed with S. cerevisiae FAS (PDB: 2uv8), P. pastoris FAS (EMD-12139), eEF3 from S. cerevisiae (EMD-12062), the S. cerevisiae ribosome (EMD-1667) with the rRNA expansion segment ES27L (PDB-3izd), the nuclear export factor Arx1 bound to the 60S large ribosomal subunit S. cerevisiae (EMD-2169), the human Ebp1 (EMD-10608), S. cerevisiae ribosomes derived from extracted ER (EMD-3764), and the ER-bound HeLa ribosomes (EMD-8056). The large subunit (LSU, 60S) of a published S. cerevisiae 80S ribosome map (EMD-3228) and the S. cerevisiae FAS map (EMD-1623) were used as references for template matching. The HeLa cell dataset is available via accession code EMD-11992.

# Field-specific reporting

Please select the one below that is the best fit for your research. If you are not sure, read the appropriate sections before making your selection.

☒ Life sciences ☐ Behavioural & social sciences ☐ Ecological, evolutionary & environmental sciences

For a reference copy of the document with all sections, see nature.com/documents/nr-reporting-summary-flat.pdf

# Life sciences study design

All studies must disclose on these points even when the disclosure is negative.

| | |
|---|---|
| Sample size | No sample size calculation was performed. From one sample of S. pombe cell culture, usually 4 cryo-grids were prepared. On each grid, around 5 lamellae with around 5 cells each could be prepared by cryo-FIB milling. On one lamella, up to 10 cryo-electron tomograms were collected. Thus, for cryo-ET sample preparation and data collection, the sample size for each experiment is sufficient. We chose 10 tomograms for training and testing. Our performance analysis showed that even less tomograms (depending on the particle to be predicted) are sufficient. Predicted particle numbers were sufficient for structural analysis, and in the case of ribosomes large enough to perform subsequent classifications. |
| Data exclusions | During sample preparation, only grid squares with several cells in the center, continuous support and sufficient ice thickness were chosen for lamella preparation. For cryo-electron tomography acquisition, areas without obstacles (e. g. crystalline ice) that are thin enough and therefore resulted in good image contrast and signal to noise ratio, were selected. Only high-quality tomograms were chosen based on thickness and residual error during tilt series alignment. |
| Replication | During sample preparation, usually 4 grids were prepared from which the ones most suitable for cryo-FIB milling were selected. The DeePiCt predictions were replicated in a robust cross-validation setting. |
| Randomization | For sample preparation, grids were prepared with cells from one cell culture condition and cryo-FIB milling performed in a non-targeted manner. For cryo-ET data collection, the selection of grids/lamellae/cells for final tomogram collection was not completely random because they were usually first selected for optimal cryo-FIB milling and areas on lamellae chosen based on ice thickness, etc. For regions meeting these quality standards that are required for high-quality cryo-ET data, the further selection process was randomized, without considerations of cytosolic content or other visible features. During structure refinement in M or RELION, particles were randomly divided into two half datasets by the software. For classification, at the particles were randomly divided evenly into the initial classes by RELION. For DeePiCt performance evaluation, the tomograms were used in cross-validation, no randomization needed. |
| Blinding | Researchers were not blinded during data collection to enable recording of cytosolic volumes which were already randomized by non-targeted cryo-FIB lamella creation. Researchers were not blinded for the analysis, and performance was evaluated by standard quality measures that were independent of subjective judgment. |

# Reporting for specific materials, systems and methods

We require information from authors about some types of materials, experimental systems and methods used in many studies. Here, indicate whether each material, system or method listed is relevant to your study. If you are not sure if a list item applies to your research, read the appropriate section before selecting a response.

## Materials & experimental systems

| n/a | Involved in the study |
|-----|----------------------|
| ☒ ☐ | Antibodies |
| ☒ ☐ | Eukaryotic cell lines |
| ☒ ☐ | Palaeontology and archaeology |
| ☒ ☐ | Animals and other organisms |
| ☒ ☐ | Human research participants |
| ☒ ☐ | Clinical data |
| ☒ ☐ | Dual use research of concern |

## Methods

| n/a | Involved in the study |
|-----|----------------------|
| ☒ ☐ | ChIP-seq |
| ☒ ☐ | Flow cytometry |
| ☒ ☐ | MRI-based neuroimaging |

