## [Peer Review File · Nature Methods]

Peer Review Information

Manuscript Title: Convolutional networks for supervised mining of molecular patterns within cellular context

Corresponding author name(s): Julia Mahamid, Judith Zaugg

Editorial Notes:

Reviewer Comments & Decisions:

Decision Letter, initial version:

31st May 2022

Dear Judith,

Your Article, "Convolutional networks for supervised mining of molecular patterns within cellular context", has now been seen by two reviewers. As you will see from their comments below, although the reviewers find your work of considerable potential interest, they have raised a number of concerns. We are interested in the possibility of publishing your paper in Nature Methods, but would like to consider your response to these concerns before we reach a final decision on publication.

We therefore invite you to revise your manuscript to address these concerns. Based on the referee feedback, we think the paper might be better suited as a Brief Communication. This would mean restructuring the paper to be 1800 words and 2 main text figures. As an alternative, if you were to add a tomographic reconstruction of a second target beyond ribosomes, you could keep the manuscript as an Article. Please let us know how you would prefer to proceed. With regards to the referee comments about filamentous structures, we think this can be discussed as a limitation or future direction, so long as it's clear the types of structures the method is best suited for. We think the other technical questions or calls for clarification should be addressed.

[Redacted] This URL links to your confidential home page and associated information about manuscripts you may have submitted, or that you are reviewing for us. If you wish to forward this email to co-authors, please delete the link to your homepage.

We hope to receive your revised paper within two months. If you cannot send it within this time, please let us know. In this event, we will still be happy to reconsider your paper at a later date so long as nothing similar has been accepted for publication at Nature Methods or published elsewhere.

OPEN SCIENCE REQUIREMENTS

REPORTING SUMMARY AND EDITORIAL POLICY CHECKLISTS

Please note that these forms are dynamic ‘smart pdfs’ and must therefore be downloaded and completed in Adobe Reader. We will then flatten them for ease of use by the reviewers. If you would like to reference the guidance text as you complete the template, please access these flattened versions at <http://www.nature.com/authors/policies/availability.html>.

DATA AVAILABILITY

All novel DNA and RNA sequencing data, protein sequences, genetic polymorphisms, linked genotype and phenotype data, gene expression data, macromolecular structures, and proteomics data must be deposited in a publicly accessible database, and accession codes and associated hyperlinks must be provided in the “Data Availability” section.

Please include a “Data availability” subsection in the Online Methods. This section should inform readers about the availability of the data used to support the conclusions of your study, including accession

codes to public repositories, references to source data that may be published alongside the paper, unique identifiers such as URLs to data repository entries, or data set DOIs, and any other statement about data availability. At a minimum, you should include the following statement: “The data that support the findings of this study are available from the corresponding author upon request”, describing which data is available upon request and mentioning any restrictions on availability. If DOIs are provided, please include these in the Reference list (authors, title, publisher (repository name), identifier, year). For more guidance on how to write this section please see:
<http://www.nature.com/authors/policies/data/data-availability-statements-data-citations.pdf>

CODE AVAILABILITY

Please include a “Code Availability” subsection in the Online Methods which details how your custom code is made available. Only in rare cases (where code is not central to the main conclusions of the paper) is the statement “available upon request” allowed (and reasons should be specified).

For more information on our code sharing policy and requirements, please see:
<https://www.nature.com/nature-research/editorial-policies/reporting-standards#availability-of-computer-code>

ORCID

Nature Methods is committed to improving transparency in authorship. As part of our efforts in this direction, we are now requesting that all authors identified as ‘corresponding author’ on published papers create and link their Open Researcher and Contributor Identifier (ORCID) with their account on the Manuscript Tracking System (MTS), prior to acceptance. This applies to primary research papers only. ORCID helps the scientific community achieve unambiguous attribution of all scholarly contributions. You can create and link your ORCID from the home page of the MTS by clicking on ‘Modify my Springer Nature account’. For more information please visit www.springernature.com/orcid.

Sincerely,
Rita

Rita Strack, Ph.D.
Senior Editor
Nature Methods

Reviewers' Comments:

Reviewer #1:

Remarks to the Author:

Review for de Teresa et al Nature Methods

This manuscript presents a method for tomogram segmentation and template matching based on convolutional neural networks showing how the algorithm efficiently detects macromolecules and sub-cellular features in tomograms of cells. The results presented are clear, and normally this would be an easy case for publication in a journal like Nature Methods, however this recommendation is complicated by the publication of another paper on the same topic (the program DeepFinder). While personally I would not be against publication, this is a decision for the editor whether it warrants publication in Nature Methods as a novel technique.

- The paper uses U-net architectures for segmentation, which is probably a good choice because it has been shown that it gives good results in almost all segmentation tasks. Although this is a sensible choice of neural networks there is no innovation from the point of computational deep learning methods. If I understood it right the only difference to DeepFinder, which uses also a U-net for segmentation is the additional network that provides cellular context.

- What I also noticed as a weakness is, that the authors did not perform an analysis with respect to the loss function of the neural networks. They only use the plain dice loss, whereas for sparse particles like FAS, more suitable loss functions (e.g. generalized dice loss, focal loss) are available which compensate for class imbalance for targets with lower abundance.

- Figure 3 is quite difficult to understand for someone not in the field, which I believe is a large readership of Nature Methods. Could the authors perhaps reduce the "jargon" and somehow in terms of percentages just compare false positives and negatives in each method?

- I think less jargon in the paper would be better. I am not sure about using terms for each dataset such as RIBO, DEF. Makes it hard work to read this paper. I am sure other readers will face similar difficulties. If the authors insist on these capitalised terms, could they provide an index somewhere?

In summary, the paper is high quality, nothing wrong with it in general. I am unsure what the advantage is over DeepFinder in terms of innovation. This is a decision for the editor whether this belongs in a more specialised journal.

Reviewer #2:

Remarks to the Author:

The manuscript reports an interesting domain-specific application of deep learning to the task of locating isolated particles in tomograms reconstructed from cryo-electron tomography data. A comprehensive training dataset of high quality has been carefully constructed and made available to the cryo-electron tomography community through EMPIAR, for which the authors should be commended. The structural results from subtomogram averaging experiments are impressive. The general approach to leveraging some knowledge about biological context inferred from a 2D CNN is elegant, as is the equalisation of spectral power for domain generalisation. Importantly, the authors also assess the ability of their solution to generalise across domains, showing for the first time that trained networks can be successfully applied to unseen species, an important demonstration. Congratulations on a fantastic piece of work.

DeePiCt is similar to the recently published DeepFinder. It does however implement additional domain-appropriate strategies for leveraging biological context and appears to perform significantly better for particles which are sparsely represented in the data. The large, curated dataset generated using the software is also extremely valuable and will prove extremely useful for the community. I am of the opinion that the segmentation-centroid approach used for particle localisation in both DeepFinder and DeePiCt severely limits their applicability to many problems for which structural cryo-ET is well suited, the study of filamentous or array-like structures.

I have been asked to comment on whether the manuscript reports a significant advance to the field and if it is likely to be of broad interest to scientists in various disciplines. Based on the data presented I believe the work reports a significant advance to the field. The software presented provides an improved, modern solution to one of the major bottlenecks in structural analysis of cryo-electron tomography data, particle localisation. This in turn enables significantly higher-throughput in structural studies of isolated proteins in situ when enough training data is available. The authors demonstrate such structural studies with impressive results from subtomogram averaging of particles localised using DeePiCt.

Please find below some specific comments and questions

- the voxel spacing in tomograms is not mentioned/discussed. This is an important parameter affecting the receptive field of a given network architecture as well as the amount of information and SNR in any given subvolume of a fixed size in pixels. I would like to see this discussed and possibly varied as a hyperparameter, with the expectation that it may significantly affect performance and applicability of the pre-trained networks to new data.

- Related to the above, some 3D data augmentation strategies were mentioned (gaussian noise, salt-and-pepper noise, rotations, elastic transformations) and said not to increase performance. Is performance measured here only as F1 score? Does augmentation change the amount of input data required to achieve a specific F1 score? This is not discussed and seems like it would be of interest even if absolute performance does not increase because prohibitively large amounts of data were required in the case of low abundance particles.

- Related: It is reported that >700 annotated instances are required for particle learning, is this independent of particle density in the tomogram?

- Architectures searched for the 3D CNN are limited to a U-Net with cross-connections, varying only the depth and the presence/absence of dropout or batch-normalisation layers. Some discussion/investigation of potential alternative architectures for segmentation (e.g. U-Nets with ResNet encoders, use of atrous convolution) would be a welcome addition.

- Discussion of alternative approaches to the problem would also be welcome, including justification of the choice to pose the problem as a segmentation problem when, as mentioned in the introduction, the task is localisation

- I understood from the methods section that subvolumes were sampled uniformly from the tomogram, unaware of regions of interest defined by the 2D CNN. I would be interested to see whether incorporating knowledge of biologically relevant regions at this stage to bias the sampling of relative subregions during training would lead to performance increases.

- I would expect that boosting the frequency with which low abundance particles are seen during training would be beneficial. Is this something that was investigated?

- The limitations of the amount of data required for the programs presented are not discussed in sufficient detail.

- Particles and coordinates can only be extracted from the centroids of segmented regions for particle localisation. This limits the direct applicability of this software to the localisation of isolated particles, ignoring in a whole class of structural cryo-ET projects for which particles are part of superstructures and cannot be expected to be localised from a centroid of a segmented region.

Author Rebuttal to Initial comments

Summary Statement:

We are grateful to the editor and the reviewers for their valuable time and inputs, and for the overall positive reviews of our work. We very much appreciate their specific questions and comments, which we have addressed in our careful revision of the manuscript. Below we provide a point-by-point response to the reviewers' comments. Significant changes to the text are highlighted in blue text in the revised manuscript documents. In brief, we now provide an example of the feasibility to address continuous filamentous assemblies by obtaining a reconstruction of a HeLa cell microtubule based on DeePiCt's segmentation. New figure panels have been incorporated into Figure 5 and Extended Data Figure 10. We further provide scripts on our GitHub repository to trace filaments from binary segmentations and derive coordinates for subsequent subtomogram analysis. Finally, we have performed a more detailed assessment of data augmentation as a strategy for reducing the amount of training data, more carefully investigated the size of training datasets needed for correct training, and evaluated the use of an additional loss function (Generalized Dice).

We hope the reviewers will now find our manuscript suitable for publication in Nature Methods.

Reviewers' Comments:**Reviewer #1:**

This manuscript presents a method for tomogram segmentation and template matching based on convolutional neural networks showing how the algorithm efficiently detects macromolecules and sub-cellular features in tomograms of cells. The results presented are clear, and normally this would be an easy case for publication in a journal like Nature Methods, however this recommendation is complicated by the publication of another paper on the same topic (the program DeepFinder). While personally I would not be against publication, this is a decision for the editor whether it warrants publication in Nature Methods as a novel technique.

Response: We thank the reviewer for the opportunity to clarify how our work deviates and advances from other methods, such as DeepFinder, in the following aspects:

Technical implementation:

- DeePiCt integrates particle localization with cellular context due to our unique implementation of 2D and 3D CNNs within the same pipeline
- DeePiCt provides a software with flexibility of architectural parameters that can be fine-tuned depending on the cellular/macromolecular structure of interest, including continuous filamentous assemblies.
- We uniquely demonstrate an implementation of transfer learning between imaging modalities (volta phase plate and defocus) and biological samples (cell type and species)

Additional resources:

- we provide trained networks that can be readily applied to any unseen tomogram
- we provide datasets with comprehensive ground truth annotations that can be used by the community to build more/better tools in the future

Demonstrate better performance for the following tested cases:

- DeePiCt models can be applied to unseen species
- DeePiCt performs better for low-density, low-abundance particle (such as FAS)
- DeePiCt can be used to segment filamentous molecular assemblies

We now highlight the differences and advances over other methods more clearly.

- The paper uses U-net architectures for segmentation, which is probably a good choice because it has been shown that it gives good results in almost all segmentation tasks. Although this is a sensible choice of neural networks there is no innovation from the point of computational deep learning methods. If I understood it right the only difference to DeepFinder, which uses also a U-net for segmentation is the additional network that provides cellular context.

Response: U-Nets are indeed a powerful tool in the field, used for a variety of tasks beyond segmentation (e.g. isonets, cryocare). We agree with the reviewer and certainly do not claim innovation on the U-Net architecture. We do claim innovation on the technical implementation, the performance of our pipeline, particularly also to segment filamentous structures, and the additional resources we provide to the community (see response to comment above).

We have now added comments on this point in the Results and Discussion sections:

- We explain why U-Nets are used in the Results: "*Here, U-Nets are a natural choice for our dual purpose of addressing segmentation and detection (the latter, achieved through post-processing steps) since they require much less parameters, and thus less training data, than architectures designed ad-hoc for object detection³²⁻³⁴.*"

- We updated the Results to point out differences with DeepFinder: "*In contrast to DeepFinder, DeePiCt's trained networks (models) for all structures mentioned in this work are open source and publicly available (Code and data availability).*"

- We updated the Discussion about performance: "*The demonstrated high performance and the flexibility of the 3D CNN architecture offer the user a reliable tool for pattern recognition. This enabled us to detect lowly abundant particle species (FAS) with a less dense structural signature compared to ribosomes, that are undetectable with other tools in the field.*"

- We added a comment about expansion of DeePiCt using additional architectures: "*As the code is open source and python-based, our flexible 3D architecture could further be expanded by implementing variations, such as ResNet encoders, atrous convolutions, class-normalization, positioning DeepPiCt to serve as a tester for deep learning techniques.*"

- We updated the Discussion to emphasize the need for high-quality ground truth data: "*A major bottleneck in the field of supervised machine learning is the availability of expert curated training data. Here, we provide the first publicly available experimental cryo-ET dataset of 20 S. pombe tomograms under two microscopy acquisition settings (VPP and defocus), together with a high-quality comprehensive annotation of ribosomes and FAS, membranes, organelle and cytosol segmentations.*"

- What I also noticed as a weakness is, that the authors did not perform an analysis with respect to the loss function of the neural networks. They only use the plain dice loss, whereas for sparse particles like FAS, more suitable loss functions (e.g. generalized dice loss, focal loss) are available which compensate for class imbalance for targets with lower abundance.

Response: Our initial tests on loss function choices showed superior results for the Dice versus Cross-Entropy loss, leading us to select the former. This choice of loss function forced us to stick to the compatible multi-label architecture, which allows multiple labels per voxel, as

opposed to multi-class neural network, which only allows a single label per voxel. Unfortunately, this implementation is neither compatible with Cross-Entropy nor Focal loss.

To specifically address the reviewer's comment, we carried out tests comparing plain Dice and Generalized Dice Loss functions. The results for this are now included in a new **Supplementary Fig. 1** (see below). We observe that Generalized Dice provides slightly superior performance than Dice in the multi-label case for ribosomes and FAS predictions, although the difference is not significant. Moreover, multi-label networks do not show significant F1-score improvement over single-class training (**Extended Data Fig. 6**). For interested users, DeePiCt provides the option to choose among these two loss functions directly from the configuration file, similar to the choice of the hyperparameters.

We have added the following paragraph in the Results section: "*Comparative analysis between DeePiCt single-class versus multi-label networks showed that single-class networks provide better results (Extended Data Fig. 6), even when loss functions that account for class imbalance, such as Generalized Dice, are employed in multi-label networks training (Supplementary Fig. 1).*"

Supplementary Figure 1 | Comparison of Dice and Generalized Dice Loss functions for a multi-label network. The results of two multi-label networks, trained with Dice and Generalized Dice Loss functions, are evaluated. Both networks were trained for simultaneous segmentation of ribosomes and FAS, two highly unbalanced classes. The results show the F1 score on 2 VPP tomograms, after being trained with 8 VPP tomograms. For the Dice Loss trained network, the results have a median F1 of 0.32 for FAS and 0.78 for ribosomes, while for the Generalized Dice Loss trained network, it has a median F1 of 0.34 for FAS and 0.81 for Ribosome.

- Figure 3 is quite difficult to understand for someone not in the field, which I believe is a large readership of Nature Methods. Could the authors perhaps reduce the "jargon" and somehow in terms of percentages just compare false positives and negatives in each method?

Response: We thank the reviewer for this suggestion and have simplified Figure 3 (see below, changes in the legend are underlined). We kept the F1 scores because it is the most comprehensive measure to report false positives and negatives. To clarify the different F1 calculations, we now added a new Supplementary Figure 2 to illustrate the performance evaluation explained in Supplementary Note 1.

Figure 3 | DeePiCt performance, cross-domain generalization and comparison with other methods. **a.** 2D CNN performance results for organelle and cytosol segmentation, when training and testing datasets are both in the same domain (VPP, cyan). The median AUPRC scores are indicated (red dashed lines). **b.** Performance of DeePiCt for the same-domain setting (VPP, cyan). The ribosome localization, FAS localization, and membrane segmentation tasks are shown (left to right). In each case, the corresponding architectures of the 3D CNN were optimized by hyperparameter tuning (Supplementary Table 1, Extended Data Fig. 6). The median F1 score for ribosomes and FAS, and a median voxel-F1 for membranes are indicated (red dashed lines). **c.** Same-domain NPC segmentation results (cyan). From left to right: performance in all, high quality, and lower quality defocus* tomograms. **d-e.** DeepFinder (d) and TM (e) particle localization results for ribosome and FAS localization. The median F1 values are indicated in each case (red dashed line). **d.** DeepFinder results were achieved by training a multiclass DeepFinder network that simultaneously segments FAS and ribosomes. **e.** TM results are shown for the VPP tomograms. **f-g.** Results of the cross-domain generalization (yellow) for the 2D CNN (f) and DeePiCt (g) (training on 10 VPP tomograms, testing on 10 defocus tomograms). Red dashed lines indicate median performance values. Different shades of grey for dots correspond to cross validation folds (DeePiCt and DeepFinder) or individual tomograms (template matching). Details about performance measurements (F1, voxel-F1, AUPRC) are described in Supplementary Note 1, Supplementary Fig. 4. **h.** Median performance summary tables associated to results in plots a-g; bold numbers denote highest score values per class.

Supplementary Figure 4 | Performance measures. **a.** A binary ground truth (blue) voxel representation and a probabilistic prediction (yellow) of a structure of interest are provided. **b.** For any given threshold $0 \leq t \leq 1$, the prediction can be binarized (yellow) and be compared to the ground truth (blue). **c.** The voxel-F1 score is the harmonic mean of voxel-based precision (P) and recall (R), computed according to the formulas in the text, considering true positive voxels as the voxels that are in the overlap (green), false positives are voxels that belong to the prediction but not to the ground truth (red), and false negatives are voxels that are in the ground truth but not in the prediction (light pink). **d.** Particle-based comparisons, on the contrary, treat each cluster of the prediction and ground truth as a separate *particle*, and thus centroids (rather than voxels) are compared. Given a radius r_0 preset by the user, a predicted particle is a true positive (green dot) when its centroid is within r_0 of a true centroid (blue dot). False positives are predicted centroids located beyond r_0 from any true centroid (red dot), and false negatives are true centroids that are located beyond r_0 from all predicted centroids (light pink dot). According to this convention, one can calculate the particle-based precision (p) and recall (r), as described in the text. **e.** The voxel-based Precision-Recall pairs ($P(t), R(t)$) parametrized by the threshold t , define the Precision-Recall Curve, whose area under the curve (AUPRC, in gray), defines a performance metric that accounts for the quality of the method across the prediction score.

- I think less jargon in the paper would be better. I am not sure about using terms for each dataset such as RIBO, DEF. Makes it hard work to read this paper. I am sure other readers will face similar difficulties. If the authors insist on these capitalised terms, could they provide an index somewhere?

Response: We have simplified the text and changed all instances of DEF to defocus, RIBO to ribosome(s), gt to ground truth, ppt to particles per tomogram. We further removed the abbreviations for the deep learning hyperparameters (D, IF, BN, ED, DD, and CV,) from the main text and now write them out (depth, initial features, batch normalisation, encoding dropout, decoding dropout, cross validation). We only keep them in the figures and supplementary notes that are directly related to parameter tuning.

In summary, the paper is high quality, nothing wrong with it in general. I am unsure what the advantage is over DeepFinder in terms of innovation. This is a decision for the editor whether this belongs in a more specialised journal.

Response: We thank the reviewer for the positive evaluation of our work and hope that now with the highlighted innovation in our response to the first point in relation to state-of-the-art tools, the reviewer will find this manuscript sufficiently novel for publication in *Nature Methods*.

Reviewer #2:

Remarks to the Author:

The manuscript reports an interesting domain-specific application of deep learning to the task of locating isolated particles in tomograms reconstructed from cryo-electron tomography data. A comprehensive training dataset of high quality has been carefully constructed and made available to the cryo-electron tomography community through EMPIAR, for which the authors should be commended. The structural results from subtomogram averaging experiments are impressive. The general approach to leveraging some knowledge about biological context inferred from a 2D CNN is elegant, as is the equalisation of spectral power for domain generalisation. Importantly, the authors also assess the ability of their solution to generalise across domains, showing for the first time that trained networks can be successfully applied to unseen species, an important demonstration. Congratulations on a fantastic piece of work.

DeePiCt is similar to the recently published DeepFinder. It does however implement additional domain-appropriate strategies for leveraging biological context and appears to perform significantly better for particles which are sparsely represented in the data. The large, curated dataset generated using the software is also extremely valuable and will prove extremely useful for the community. I am of the opinion that the segmentation-centroid approach used for particle localisation in both DeepFinder and DeePiCt severely limits their applicability to many problems for which structural cryo-ET is well suited, the study of filamentous or array-like structures.

Response: We agree with the reviewers about the importance segmenting filamentous structures. DeePiCt is designed such that users can choose between centroids (for particles) and segmentation (for continuous structures such as filaments and membranes). Thus indeed, DeePiCt can already do what the reviewer suggests. To showcase this better, we have now added a reconstruction of a HeLa cell microtubule structure based on DeePiCt segmentations, shown in revised Figure 5 and Extended Data Figure 10. We now also provide scripts for performing downstream analysis on filament segmentations towards structural analyses on the DeePiCt GitHub repository. These include filament tracing and coordinate sampling required for the reconstruction of the subtomogram average.

Figure 5 | DeePiCt's generalization across species. A dataset depicting a HeLa cell nuclear periphery¹ is segmented by applying four independently trained DeePiCt networks. The results show the segmentation of actin filaments (red) trained on RPE-1 and MEF 3T3 tomograms, microtubules (MTs, cyan) trained on *C. elegans* tomograms, and cytosolic ribosomes (yellow) and membranes (purple) trained on *S. pombe* tomograms. The inset (top right corner) in the DeePiCt predictions panel shows the MT subtomogram average (cyan) obtained from the displayed segmentations.

I have been asked to comment on whether the manuscript reports a significant advance to the field and if it is likely to be of broad interest to scientists in various disciplines. Based on the data presented I believe the work reports a significant advance to the field. The software presented provides an improved, modern solution to one of the major bottlenecks in structural analysis of cryo-electron tomography data, particle localisation. This in turn enables significantly higher-throughput in structural studies of isolated proteins in situ when enough training data is available. The authors demonstrate such structural studies with impressive results from subtomogram averaging of particles localised using DeePICT.

Response: Thank you for your positive evaluation of the manuscript.

Please find below some specific comments and questions

- the voxel spacing in tomograms is not mentioned/discussed. This is an important parameter affecting the receptive field of a given network architecture as well as the amount of information and SNR in any given subvolume of a fixed size in pixels. I would like to see this discussed and possibly varied as a hyperparameter, with the expectation that it may significantly affect performance and applicability of the pre-trained networks to new data.

Response: We thank the reviewer for pointing out this shortcoming in the presentation of our work. We chose to perform the final training and predictions at a common binning of 4x for the tomogram reconstruction, providing a pixel spacing in the range of 10 Å. This is indeed a parameter that each user can choose for their reconstruction. We have performed initial trials on using different binning, especially for the challenging case of FAS. We observed that 2 times binned tomograms did not allow the network to learn the FAS structure from the data. But we did not specifically test if this was due to lower SNR or to a smaller receptive field of the network. A test would imply using 2 times binned tomograms and deeper networks (depth=3), which in turn would require more trainable parameters to be learnt; the tests can be carried by interested users with our code.

- We added a paragraph about the choice of voxel spacing in the Results: "*The physical receptive field of the network, i.e. the extent of the context it is able to see for each prediction voxel, depends not only on the depth hyperparameter, but also on the physical inter-voxel spacing. In all experiments we used 4 times binned tomograms with inter-voxel spacing of ~13.5 Å (Online Methods), sufficient for the detection of ~30 nm diameter particles investigated here. The binning can, and in some cases should, be optimized to provide adequate SNR while retaining sufficient information to detect structures of interest. Lower binning retains higher resolution information at the cost of higher noise and also decreases the physical receptive field of the network. The latter could be compensated with a higher depth, thus requiring more training data and possibly amplifying the well-known vanishing gradient effect, associated to deeper architectures. Higher binning may be sufficient for larger structures such as organelles.*"

- Related to the above, some 3D data augmentation strategies were mentioned (gaussian noise, salt-and-pepper noise, rotations, elastic transformations) and said not to increase performance. Is performance measured here only as F1 score? Does augmentation change the amount of input data required to achieve a specific F1 score? This is not discussed and

seems like it would be of interest even if absolute performance does not increase because prohibitively large amounts of data were required in the case of low abundance particles.

Response: We have performed the analysis suggested by the reviewer to address this and the next point (aspects of data augmentation and training set size). The results are now added to a new section in Supplementary Note 1, *Training set sizes for particle picking*, where we included new **Supplementary Figure 4** (see below).

Specifically, we trained networks on 1, 3, 5, and 8 tomograms for ribosomes, and on 2, 4, 6 and 8 tomograms for FAS. In both cases we trained and evaluated with and without data augmentation and then compared the F1 scores. We found that data augmentation did not improve results significantly (see new **Supplementary Figure 2**).

Supplementary Figure 2 | Performance with and without data augmentation across training set sizes. a-d. Plots show network prediction performance on two tomograms when varying training set sizes for ribosome and FAS, without (- AUG, a and c) and with (+ AUG, b and d) data augmentation. For ribosomes (a-b), the naturally high numbers of particles per tomogram allow high F1 scores even when training on one tomogram (2,450 particles). FAS results (c-d) show that 300 particles are enough to get a fully trained network. In both cases data augmentation strategies including additive Gaussian noise, salt-and-pepper noise, and random rotations around the z-axis do not show clear advantages (b, d).

- Related: It is reported that >700 annotated instances are required for particle learning, is this independent of particle density in the tomogram?

Response: We further investigated the required training set sizes. Both particle species tested here are overall evenly distributed in the cellular volumes, with ribosomes being frequent and FAS rather sparse. We show that for ribosomes a single tomogram (~2000 particles) is sufficient to achieve maximum F1, while for FAS ~300 particles from 2 tomograms were required. Our experiments suggest that learning occurs independently of particle density.

We have added this result to the Results section and updated the number 700 to 300: *"Our tests on experimental data show that about 5 tomograms are sufficient for membrane segmentation, while a minimum of 300 annotated instances is required for particle learning in the 3D CNNs (Supplementary Note 1, Supplementary Fig. 2). Learning was observed to be independent of how sparse particles are in the cellular volumes."*

- Architectures searched for the 3D CNN are limited to a U-Net with cross-connections, varying only the depth and the presence/absence of dropout or batch-normalisation layers. Some discussion/investigation of potential alternative architectures for segmentation (e.g. U-Nets with ResNet encoders, use of atrous convolution) would be a welcome addition.

Response: We chose a U-Net architecture because of its proven success in a variety of tasks and its advantage compared to detection networks (less training data required). The number of all possible variations on the architecture to be utilized is very large and we consider this outside of the scope for the current manuscript. However, we do expect the community to pick up on this, potentially using DeePiCt with its flexible architecture as a starting point.

- We have added a motivation for using U-Nets in the Results: *"U-Net architectures have become the standard for much of modern deep learning, with enormous success beyond segmentation, including in denoising³⁰ and reconstruction³¹ methods for cryo-ET. Here, U-Nets are a natural choice for our dual purpose of addressing segmentation and detection (the latter, achieved through post-processing steps) since they require much less parameters, and thus less training data, than architectures designed ad-hoc for object detection³²⁻³⁴"*

- And a point about additional architectures to the Discussion: *"As the code is open source and python-based, our flexible 3D architecture could further be expanded by implementing variations, such as ResNet encoders, atrous convolutions, class-normalization, positioning DeePiCt to serve as a tester for deep learning techniques."*

- Discussion of alternative approaches to the problem would also be welcome, including justification of the choice to pose the problem as a segmentation problem when, as mentioned in the introduction, the task is localization.

Response: The task of mining different structures in cryo-electron tomograms requires both segmentation and localization, depending on the structure of interest and the desired downstream processing. The advantage of a segmentation network is that it requires much fewer trainable parameters than object detection networks (which are characterized by a heavier architecture, especially in the case of 3D CNNs), and it can still perform object detection (localization) by extracting coordinates in the post-processing stage (please see comment immediately above).

We have added a comment Results section (same as above): *"Here, U-Nets are a natural choice for our dual purpose of addressing segmentation and detection (the latter, achieved through post-processing steps), since they require much less parameters, and thus smaller training sets, than architectures designed for ad-hoc for object detection³²⁻³⁴."*

- I understood from the methods section that subvolumes were sampled uniformly from the tomogram, unaware of regions of interest defined by the 2D CNN. I would be interested to see

whether incorporating knowledge of biologically relevant regions at this stage to bias the sampling of relative subregions during training would lead to performance increases.

Response: We agree this is an interesting question for future investigations by testing whether region masks make a difference already at the training stage. In our method, the use of region masks is implemented at the post-processing stage to select structures such as ribosomes in certain areas of the tomogram (e.g. close to ER and mitochondria). Users of the software can use these filtered predictions to train new models for specific use cases. However, this approach could be limited by the amount of training data and thus has to be tailored to a specific biological question.

- I would expect that boosting the frequency with which low abundance particles are seen during training would be beneficial. Is this something that was investigated?

Response: The frequency of low abundance particles can be increased during training by incorporating only boxes with positive labels. While we have not tested this option, DeePiCt offers the possibility for the user to choose boxes that have more than a chosen percentage of labels by adjusting the "min label fraction".

We have now clarified this point in the Supplementary Note 1: *"The presence and the frequency of the structure in the training set is one of the parameters that in our experience largely determines whether the 3D CNN is able to learn. The user can set in the configuration file a minimum label presence ($0 \leq \text{min_label_fraction} \leq 1$), which indicates the proportion of labeled voxels in the patch that is required for it to be considered among the training patches. In this way, patches with too few labeled voxels are eliminated."*

- The limitations of the amount of data required for the programs presented are not discussed in sufficient detail.

Response: We agree with this and refer the reviewer to our response to a comment above.

Briefly, we have added Supplementary Figure 2 addressing this question and now emphasize this aspect in the Results: *"Our tests on experimental data show that about 5 tomograms are sufficient for membrane segmentation, while a minimum of 300 annotated instances is required for particle learning in the 3D CNNs (Supplementary Note 1, Supplementary Fig. 2). Learning was observed to be independent of how sparse particles are in the cellular volumes."*

- Particles and coordinates can only be extracted from the centroids of segmented regions for particle localisation. This limits the direct applicability of this software to the localisation of isolated particles, ignoring in a whole class of structural cryo-ET projects for which particles are part of superstructures and cannot be expected to be localised from a centroid of a segmented region.

Response: We agree with the reviewer about the importance of localizing superstructures. And have responded to this in response to the first point of this reviewer (copied here again for convenience):

DeePiCt is designed such that users can choose between centroids (for particles) and segmentation (for continuous structures such as filaments and membranes). Thus indeed, DeePiCt can already do what the reviewer suggests. To showcase this better, we have now

added a reconstruction of a HeLa cell microtubule structure based on DeePiCt segmentations, shown in revised **Figure 5** and **Extended Data Figure 10**. We now also provide scripts for performing downstream analysis on filament segmentations towards structural analyses on the DeePiCt GitHub repository. These include filament tracing and coordinate sampling required for the reconstruction of the subtomogram average.

Further to this, we emphasize the reconstruction of a filamentous structures in the Results: "*The voxel-based DeePiCt prediction can be used to trace filaments and derive 3D coordinates for subsequent subtomogram averaging of microtubules (Online Methods). This analysis revealed the HeLa cell microtubules' hollow structure with 13 protofilaments and a diameter of 25 nm (Fig. 5, Extended Data Fig. 10 c)*"

Figure 5 | DeePiCt's generalization across species. A dataset depicting a HeLa cell nuclear periphery¹ is segmented by applying four independently trained DeePiCt networks. The results show the segmentation of actin filaments (red) trained on RPE-1 and MEF 3T3 tomograms, microtubules (MTs, cyan) trained on *C. elegans* tomograms, and cytosolic ribosomes (yellow) and membranes (purple) trained on *S. pombe* tomograms. The inset (top right corner) in the DeePiCt predictions panel shows the MT subtomogram average (cyan) obtained from the displayed segmentations.

Extended Data Fig. 10 | Domain generalization of different cellular structures and species. [...].c. Predictions in the HeLa cell dataset (left) of MT model trained with *C. elegans* tomograms (voxel-F1=0.83). Coordinates (middle) were extracted from the binary prediction map (left) using custom scripts (Code and data availability). The inset shows a zoom into the boxed area (white box). A 3D microtubule average (right, cyan) at 39 Å resolution was generated using Dynamo⁴⁰. 2D slices through the volume show the xy plane (top) with the individual microtubule protofilaments labeled, and the xz plane (bottom) with the 25 nm microtubule and 17 nm lumen diameter indicated.

Decision Letter, first revision:

Dear Judith,

Thank you for submitting your revised manuscript "Convolutional networks for supervised mining of molecular patterns within cellular context" (NMETH-A48889A). It has now been seen by the original referees and their comments are below. The reviewers find that the paper has improved in revision, and

therefore we'll be happy in principle to publish it in Nature Methods, pending minor revisions to comply with our editorial and formatting guidelines.

TRANSPARENT PEER REVIEW

Nature Methods offers a transparent peer review option for new original research manuscripts submitted from 17th February 2021. We encourage increased transparency in peer review by publishing the reviewer comments, author rebuttal letters and editorial decision letters if the authors agree. Such peer review material is made available as a supplementary peer review file. Please state in the cover letter 'I wish to participate in transparent peer review' if you want to opt in, or 'I do not wish to participate in transparent peer review' if you don't. Failure to state your preference will result in delays in accepting your manuscript for publication.

Thank you again for your interest in Nature Methods Please do not hesitate to contact me if you have any questions.

Sincerely,
Rita

Rita Strack, Ph.D.
Senior Editor
Nature Methods

ORCID

Reviewer #1 (Remarks to the Author):

The authors have addressed all my comments satisfactorily. I have no further criticisms.

Reviewer #2 (Remarks to the Author):

All concerns raised have been satisfactorily addressed, thank you to the authors for the time taken to perform extra analyses based on the previous review.

Kind regards,

Alister Burt

Final Decision Letter:

Dear Judith,

Here I am sending the formal accept decision. I see in our notes that we do have one remaining request. Can you please email me a version of your Supplementary Information that has the table of contents removed as soon as is convenient? We need to have this version to export your paper to production.

I am pleased to inform you that your Article, "Convolutional networks for supervised mining of molecular patterns within cellular context", has now been accepted for publication in Nature Methods. Your paper is tentatively scheduled for publication in our * print issue, and will be published online prior to that. The received and accepted dates will be April 12, 2022 and Dec 2, 2022. This note is intended to let you know what to expect from us over the next month or so, and to let you know where to address any further questions.

Over the next few weeks, your paper will be copyedited to ensure that it conforms to Nature Methods style. Once your paper is typeset, you will receive an email with a link to choose the appropriate publishing options for your paper and our Author Services team will be in touch regarding any additional information that may be required.

Your paper will now be copyedited to ensure that it conforms to Nature Methods style. Once proofs are generated, they will be sent to you electronically and you will be asked to send a corrected version within 24 hours. It is extremely important that you let us know now whether you will be difficult to contact over the next month. If this is the case, we ask that you send us the contact information (email, phone and fax) of someone who will be able to check the proofs and deal with any last-minute problems.

If, when you receive your proof, you cannot meet the deadline, please inform us at rjsproduction@springernature.com immediately.

Once your manuscript is typeset and you have completed the appropriate grant of rights, you will receive a link to your electronic proof via email with a request to make any corrections within 48 hours. If, when you receive your proof, you cannot meet this deadline, please inform us at rjsproduction@springernature.com immediately.

Once your paper has been scheduled for online publication, the Nature press office will be in touch to confirm the details.

Content is published online weekly on Mondays and Thursdays, and the embargo is set at 16:00 London time (GMT)/11:00 am US Eastern time (EST) on the day of publication. If you need to know the exact publication date or when the news embargo will be lifted, please contact our press office after you have submitted your proof corrections. Now is the time to inform your Public Relations or Press Office about your paper, as they might be interested in promoting its publication. This will allow them time to prepare an accurate and satisfactory press release. Include your manuscript tracking number NMETH-A48889B and the name of the journal, which they will need when they contact our office.

About one week before your paper is published online, we shall be distributing a press release to news organizations worldwide, which may include details of your work. We are happy for your institution or funding agency to prepare its own press release, but it must mention the embargo date and Nature Methods. Our Press Office will contact you closer to the time of publication, but if you or your Press Office have any inquiries in the meantime, please contact press@nature.com.

If you are active on Twitter, please e-mail me your and your coauthors' Twitter handles so that we may tag you when the paper is published.

Please note that *Nature Methods* is a Transformative Journal (TJ). Authors may publish their research with us through the traditional subscription access route or make their paper immediately open access through payment of an article-processing charge (APC). Authors will not be required to make a final decision about access to their article until it has been accepted. [Find out more about Transformative Journals](https://www.springernature.com/gp/open-research/transformative-journals)

To assist our authors in disseminating their research to the broader community, our SharedIt initiative provides you with a unique shareable link that will allow anyone (with or without a subscription) to read the published article. Recipients of the link with a subscription will also be able to download and print the PDF. As soon as your article is published, you will receive an automated email with your shareable link.

Please note that you and your coauthors may order reprints and single copies of the issue containing your article through Nature Portfolio's reprint website, which is located at <http://www.nature.com/reprints/author-reprints.html>. If there are any questions about reprints please send an email to author-reprints@nature.com and someone will assist you.

Best regards,

Rita